# EmoPrefer: Can Large Language Models Understand Human Emotion Preferences?

**Zheng Lian**[1,2*]**, Licai Sun**[3]**, Lan Chen**[4]**, Haoyu Chen**[3]**, Zebang Cheng**[5,6]**, Fan Zhang**[7]**,
Ziyu Jia**[4]**, Ziyang Ma**[8]**, Fei Ma**[6]**, Xiaojiang Peng**[9]**, Jianhua Tao**[10*]

[1]State Key Laboratory of Autonomous Intelligent Unmanned Systems, Tongji University
[2]Frontiers Science Center for Intelligent Autonomous Systems, Ministry of Education
[3]University of Oulu [4]MAIS, CASIA [5]Shenzhen University
[6]Guangdong Laboratory of Artificial Intelligence and Digital Economy (SZ)
[7]The Chinese University of Hong Kong [8]Shanghai Jiaotong University
[9]Shenzhen Technology University [10]Department of Automation, BNRist, Tsinghua University

## Abstract

Descriptive Multimodal Emotion Recognition (DMER) has garnered increasing research attention. Unlike traditional discriminative paradigms that rely on predefined emotion taxonomies, DMER aims to describe human emotional state using free-form natural language, enabling finer-grained and more interpretable emotion representations. However, this free-form prediction paradigm introduces new challenges regarding its evaluation. Previous works depend on ground-truth descriptions, but emotions are inherently tied to diverse human behaviors, and generating a comprehensive and accurate description is inherently demanding. Other researchers reformulate this problem into a more tractable human preference learning task, but pairwise preference annotation involves substantial manual effort. This leads to a question: *can we leverage multimodal LLMs (MLLMs) to achieve more cost-efficient preference annotation?* To answer this, we propose **EmoPrefer**, a pioneering work exploring the potential of LLMs in decoding human emotion preferences. Specifically, we construct the first emotion preference dataset, **EmoPrefer-Data**, featuring high-quality preference annotations from experts. Additionally, we introduce **EmoPrefer-Bench**, which evaluates the performance of various MLLMs and prompting techniques in preference prediction, while also revealing new strategies to enhance their performance. To the best of our knowledge, this is the first work exploring the capabilities of LLMs in understanding human emotion preferences. Our work advances the field of DMER and lays the foundation for more intelligent human-computer interaction. Our data and code are released at https://github.com/zeroQiaoba/AffectGPT/tree/master/EmoPrefer.

## 1 Introduction

Descriptive Multimodal Emotion Recognition (DMER) (Lian et al., 2023; 2025a) aims to use free-form natural language to describe emotional states. Unlike traditional *discriminative* methods that depend on predefined emotion taxonomies (El Ayadi et al., 2011; Lian et al., 2026), DMER adopts a *generative* paradigm, offering greater flexibility in emotion representation. This enables fine-grained and interpretable emotional expression, presenting significant opportunities for advancing emotion-intelligent human-computer interaction technologies (Brave & Nass, 2007). Recent advancements in Multimodal Large Language Models (MLLMs) (Zhao et al., 2023; Yin et al., 2024), with their rich vocabularies and multimodal understanding capabilities, have made this task feasible. Figure 1 demonstrates the distinctions between descriptive and discriminative MER.

However, evaluating the quality of such open-ended descriptions remains a non-trivial task. Figure 2 summarizes the existing evaluation strategies. The first approach leverages ground-truth emotion descriptions and employs LLMs, such as OpenAI GPT (OpenAI, 2024) or Google Gemini (Team et al., 2023), to measure the similarity between predicted and ground-truth descriptions (Lian et al.,

---

*Corresponding Author

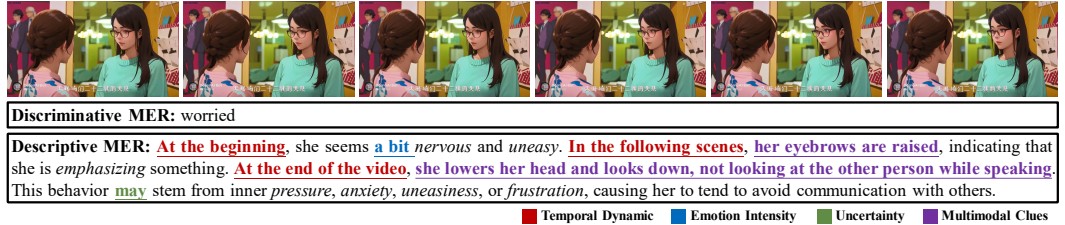

Figure 1: **Task comparison.** *Discriminative MER* assigns a label from a predefined taxonomy, whereas *Descriptive MER* provides a more detailed description by incorporating emotional temporal dynamics, intensity, uncertainty, and the existence and reasonableness of multimodal clues.

2023). Nevertheless, emotions are inherently tied to diverse human behaviors, including facial expressions, (micro-)gestures, head movements, hand actions, and vocal tones (Chen et al., 2023; Li & Deng, 2020; Yang et al., 2026). As a result, generating a comprehensive and accurate description of a person's emotional state is inherently challenging, and unreliable ground-truth descriptions can lead to inaccurate evaluations. To address this, researchers propose abandoning costly and often incomplete ground-truth descriptions (Lian et al., 2025b). Instead, they reformulate the complex problem of designing indicators to measure semantic similarity with ground-truth descriptions into a more tractable problem of learning human preferences. However, this approach requires preference annotations for each model pair across multiple samples, entailing substantial human effort. For example, given $M$ models and $N$ samples, the total number of required comparisons is $C(M, 2) \times N$.

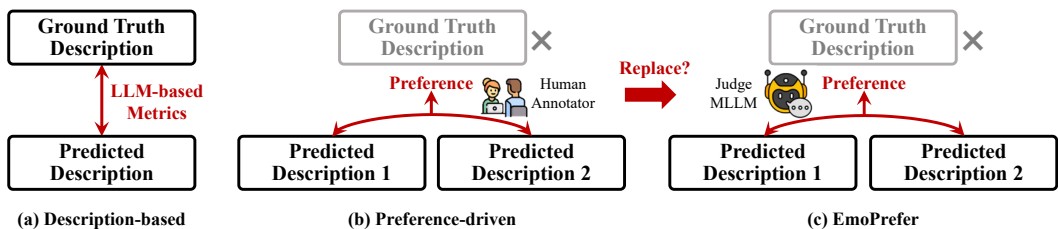

Figure 2: **Evaluation strategies for DMER.** (a) *Description-based evaluation*, which relies on costly and often incomplete human-annotated descriptions; (b) *Preference-driven evaluation*, where pairwise preference annotations require substantial human effort; (c) *EmoPrefer*, a pioneering approach that explores whether MLLMs can decode human emotion preferences, aiming to replace expensive human annotations with MLLM-based judges.

Given the high cost of manual preference annotation, a heuristic idea emerges: *Can we leverage MLLMs to achieve automatic emotion preference decoding, thereby providing a more cost-efficient evaluation strategy for DMER?* To answer this, we propose **EmoPrefer**, the first work exploring the potential of MLLMs in emotion preference prediction. (1) We introduce **EmoPrefer-Data**, the first multimodal preference dataset centered on human emotions. In this dataset, we provide pairwise emotion descriptions for videos and recruit multiple expert annotators to label preferences. Only samples with unanimous agreement among all annotators are retained, ensuring high-quality preference annotations. (2) We establish **EmoPrefer-Bench**, the first benchmark for emotion preference prediction. In this benchmark, we conduct a comprehensive evaluation of different MLLMs and prompting techniques, further exploring strategies to enhance their alignment with human preferences. This paper presents pioneering work that reveals the capabilities of MLLMs in human emotion preference. Beyond evaluation, the dataset and insights from our benchmark will support the training of emotion-aware reward models, thereby enhancing the consistency of MLLMs with human emotional understanding. The main contributions of this paper are summarized as follows:

- **(EmoPrefer)** This is a pioneering work that explores the potential of MLLMs in emotion preference. Beyond evaluation, the insights from this paper will contribute to training reward models capable of understanding human emotions, paving the way for emotion-intelligent models.

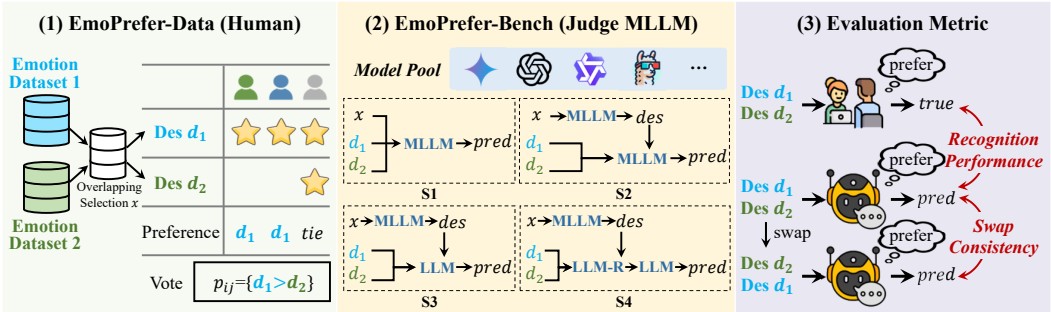

Figure 3: **Pipeline of EmoPrefer.** (1) EmoPrefer-Data. We select overlapping videos across two descriptive emotion datasets, then conduct human preference annotations and adopt their majority-voted results as final labels. (2) EmoPrefer-Bench. We systematically evaluate various MLLMs and prompting techniques, revealing their potential in emotion preference decoding. (3) Evaluation metric. We introduce two metrics for performance evaluation: *recognition performance* measures alignment between MLLM-based judges and human annotations; *swap consistency* evaluates the MLLM-based judges' robustness to swapping the order of description pairs.

- **(EmoPrefer-Data)** This paper constructs the first preference dataset centered on emotions. Our work makes a substantial contribution to current research on LLMs-as-judges by not only expanding the modality to full multimodal scenarios (encompassing audio, video, and text) but also extending preference learning to the domain of human emotions.

- **(EmoPrefer-Bench)** This paper proposes comprehensive evaluation metrics and explores diverse solutions, demonstrating the feasibility of automatic preference prediction for human emotions. Additionally, we introduce model-based crowdsourcing to enhance performance.

## 2 EMOPREFER-DATA

Figure 3 illustrates the overall pipeline of **EmoPrefer**, which is the first work to explore the potential of MLLMs for emotion preference decoding. To achieve this, we require a high-quality, human-annotated emotion preference dataset. In this section, we detail the construction process of our **EmoPrefer-Data** from four aspects: (1) sample selection, (2) description quality analysis, (3) preference annotation, and (4) inter-annotator agreement analysis.

**Sample Selection.** For preference annotation, we provide pairwise emotion descriptions for each video and ask human annotators to select the better one. However, if the quality of two descriptions varies significantly, the preference prediction task will become trivial, potentially diminishing the advantages of superior solutions. To ensure high-quality descriptions, we leverage two benchmark datasets for descriptive emotion: MERR-Fine (Cheng et al., 2024) and MER-Caption+ (Lian et al., 2025a). The raw videos in these datasets are sourced from the MER2024 dataset (Lian et al., 2024). We only select videos that appear in both datasets, resulting in 1,368 samples, each with two descriptions (one from each dataset). Appendix C.1 provides visualizations of the selected video samples. These videos primarily feature a single front-facing character with complete audio content, ensuring the full expression of emotions for the target speaker.

**Description Quality Analysis.** This section analyzes the descriptions provided by MERR-Fine and MER-Caption+. These datasets focus on capturing the character's multimodal cues in the video, including facial expressions, body language, vocal tone, contextual events, and environmental factors, offering a comprehensive depiction of the person's emotional state. Figure 4 presents word clouds of emotion words and nouns extracted from these descriptions. The detailed extraction process is described in Appendix C.2. Our analysis reveals that these descriptions contain a rich diversity of emotional vocabulary and multimodal cues related to emotions, confirming their high quality.

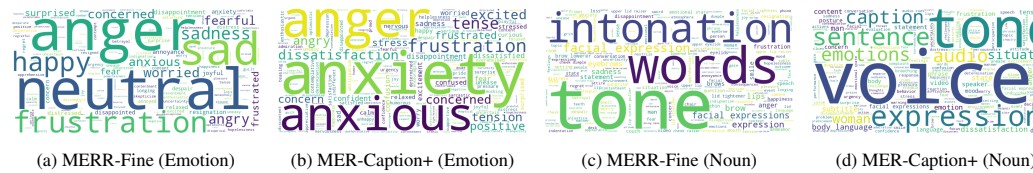

| (a) MERR-Fine (Emotion) | (b) MER-Caption+ (Emotion) | (c) MERR-Fine (Noun) | (d) MER-Caption+ (Noun) |

Figure 4: **Word cloud visualization.** We extract emotion words and nouns from the descriptions and generate corresponding word clouds for both datasets. Appendix C.2 details the extraction process.

**Preference Annotation.** During the annotation process, we recruit master's students from our lab as candidate annotators. Given their research focus on affective computing, these individuals are already familiar with emotion definitions. To ensure their understanding aligns with that of the general population, we first conduct a preliminary test. Specifically, we select 12 samples that have reached consensus among eight external annotators (i.e., all eight have assigned identical emotion labels). The candidates are then asked to annotate these 12 samples, and only those who achieve at least 75% accuracy are retained, resulting in three qualified annotators. The selected annotators are then tasked with determining which of two descriptions more accurately reflects the character's emotional state. Appendix C.3 illustrates the layout of our annotation platform. For each comparison, annotators can choose one of three options: *the first description is better*, *the second description is better*, or *tie*. We retain only samples where all three annotators reach consensus. This strict selection strategy ensures the reliability of the preference labels.

**Inter-annotator Agreement Analysis.** After annotation, we evaluate the inter-annotator agreement for emotion preference recognition. Figure 5a presents the agreement scores excluding ties, with an average of $\mathrm{avg}(69.88, 69.31, 68.73) = 69.31\%$. This reflects the upper bound of human agreement and confirms that different annotators can reach a reasonable consensus on this task. Figure 5b shows the agreement scores including ties, where the average drops to $\mathrm{avg}(60.87, 56.39, 60.43) = 59.23\%$, a 10% decrease compared to Figure 5a. This suggests that annotators struggle more to reach consensus on ties, indicating that ties represent higher ambiguity than clear preferences (e.g., when one description is distinctly better than another). Table 1 compares EmoPrefer-Data with existing datasets. Our dataset not only covers more modalities but also extends the preference task to human emotions. To the best of our knowledge, this is the first multimodal preference dataset centered on human emotions, providing valuable resources for future research.

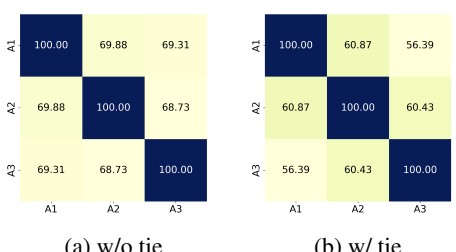

| (a) w/o tie | (b) w/ tie |

Figure 5: **Inter-annotator agreement analysis.** We present the inter-annotator agreement under two conditions: w/o ties and w/ ties.

## 3 EVALUATION METRICS

In this section, we propose two metrics to evaluate the performance of MLLM-based judges: *recognition performance* and *swap consistency*. Figure 3 illustrates the detailed calculation pipeline.

**Recognition Performance.** This metric evaluates the alignment between model predictions and human annotations. In EmoPrefer-Data, the labels fall into three categories: (1) *the first description is better*, (2) *the second description is better*, or (3) *tie*. Since human annotators often struggle to reach consensus on ties (as shown in Figure 5), we report both the weighted average F1-score (WAF) and accuracy (ACC) under two settings: two-class (excluding ties) and three-class (including ties) setups. Given the inherent class imbalance in the dataset, we default to the two-class WAF when reporting a single metric in the following experiments.

**Swap Consistency.** Ideally, swapping the order of description pairs should not affect the model's preferred description. However, in practice, we observe that order swaps sometimes lead to different

Table 1: **Dataset comparison.** The second column specifies the modalities used for preference tasks. Since MLLM-as-a-Judge (Chen et al., 2024) relies on image sequences rather than videos, we use the symbol ✔ to indicate this. Our EmoPrefer-Data not only expands the range of modalities but also extends the preference task to human emotions.

| Dataset | Modality | | | | Task Description |
| --- | T | I | A | V | --- |
| MT-Bench (Zheng et al., 2023) | ✔ | ✗ | ✗ | ✗ | writing, math, general knowledge, etc. |
| LLMEval (Zhang et al., 2023) | ✔ | ✗ | ✗ | ✗ | storytelling, summarization, dialogue, etc. |
| FairEval (Wang et al., 2024a) | ✔ | ✗ | ✗ | ✗ | writing, role play, etc. |
| PandaLM (Wang et al., 2024b) | ✔ | ✗ | ✗ | ✗ | law, biomedical data, etc. |
| JudgeBench (Tan et al., 2025) | ✔ | ✗ | ✗ | ✗ | general knowledge, reasoning, math, coding |
| CALM (Ye et al., 2025) | ✔ | ✗ | ✗ | ✗ | commonsense, math, science, etc. |
| MLLM-as-a-Judge (Chen et al., 2024) | ✔ | ✔ | ✗ | ✔ | captioning, reasoning, etc. |
| VL-RewardBench (Li et al., 2025c) | ✔ | ✔ | ✗ | ✗ | general visual perception, reasoning, etc. |
| MM-RLHF (Zhang et al., 2025) | ✔ | ✔ | ✗ | ✔ | captions, safety, reasoning, etc. |
| Multimodal RewardBench (Yasunaga et al., 2025) | ✔ | ✔ | ✗ | ✗ | knowledge, safety, reasoning, etc. |
| **EmoPrefer-Data (Ours)** | ✔ | ✔ | ✔ | ✔ | human emotion understanding |

outcomes. This discrepancy may arise because the model's preferences are determined by the order of inputs rather than the actual content of the descriptions. Therefore, we introduce *swap consistency*, a metric that evaluates a model's robustness to order swapping and ensures predictions are based on content rather than ordering. Specifically, given $N$ examples $\{(x^n, d_1^n, d_2^n)\}_{i=1}^N$, where $x^n$ is a video and $d_1^n$ and $d_2^n$ are two corresponding descriptions, we use MLLM to predict preferences for both the normal order input $(x^n, d_1^n, d_2^n)$ and the swapped order input $(x^n, d_2^n, d_1^n)$. This yields two outcomes: $o_{12}^n$ and $o_{21}^n$. We compute swap consistency as follows:

$$\text{Swap Consistency} = \frac{\sum_{i=1}^N \mathbb{I}(o_{12}^n = o_{21}^n)}{N}, \tag{1}$$

where $\mathbb{I}(\cdot)$ is the indicator function, and higher scores indicate greater robustness to input order variations. Appendix E provides the prompt template used in the calculation process.

## 4 EMOPREFER-BENCH

This is the first benchmark designed for emotion preference recognition. We evaluate various MLLMs and explore the effectiveness of model-based crowdsourcing. The paper reports the zero-shot performance of these MLLMs, with all experiments conducted on A100 GPUs. To reduce randomness, each experiment is run twice, and both the mean and standard deviation are reported.

### 4.1 JUDGE MLLM

Given that humans express emotions through multiple modalities, we require MLLMs to support at least audio or video inputs. To enable preference prediction, we propose four prompting strategies (see Figure 3). **Strategy 1 (S1):** We input the video along with two descriptions into the MLLM and ask the model to determine the better one based on the character's emotional state. **Strategy 2 (S2):** We decompose S1 into two steps. First, we instruct the MLLM to generate a detailed description of the video. Then, using the output from the first step as a reference, we prompt the model to determine which of the two descriptions better aligns with it. **Strategy 3 (S3):** The second step in S2 relies solely on text input, meaning it can be implemented using an external LLM. The rationale for this alternative is that current MLLM training typically begins with a pretrained LLM, followed by additional multimodal fine-tuning. While this enables the model to process other modalities, such training may compromise its text understanding capabilities. Thus, we opt to use an external LLM for the second step. **Strategy 4 (S4):** We further break down the second step in S3 into two sub-steps, requiring the model to perform additional reasoning before making a decision. This allows us to investigate whether the reasoning process is necessary for accurate preference prediction. We adopt Qwen2.5-7B (Yang et al., 2024) as the external LLM for S3/S4, and the rationale for this choice is discussed in Section 4.2. Appendix D provides the prompt templates for all strategies.

## 4.2 MAIN RESULTS

**Impact of Prompting Strategy.** Section 4.1 proposes four prompting strategies, and their impacts on different MLLMs are summarized in Figure 6. For most models, S3 and S4 outperform S1 and S2. The key distinction between these strategies lies in whether they leverage an external LLM. These findings highlight the advantages of integrating an external LLM, as MLLMs may have limitations in language understanding. However, some exceptions exist. For instance, Qwen2.5-VL (Bai et al., 2025) and Qwen2.5-Omni (Xu et al., 2025) perform better with S1 and S2 than with S3 and S4. This is because these models retain strong language processing capabilities due to their advanced training methodologies, enabling them to interpret complex prompts effectively. In such cases, the longer inference chains in S3 and S4 may introduce error accumulation and degrade performance. Thus, a trade-off emerges: *while using an external LLM can compensate for the limited language processing abilities of some MLLMs, longer inference chains may also increase error accumulation.*

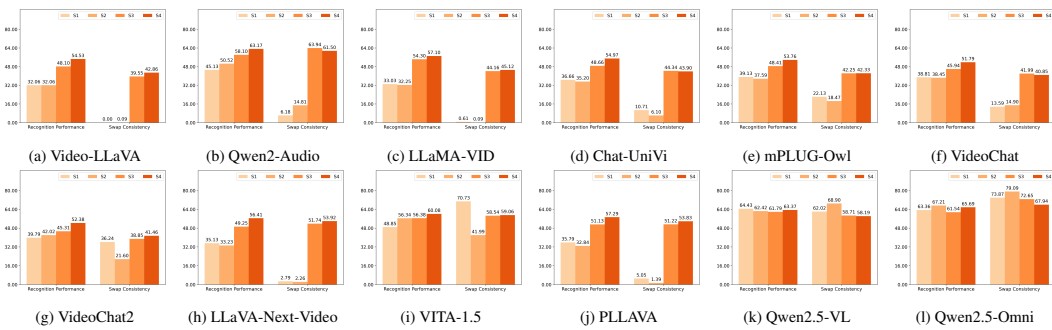

Figure 6: **Impact of prompting strategies.** In these figures, we compare the performance of S1∼S4 across various MLLMs. All experiments are conducted on the EmoPrefer-Data dataset.

**Role of External LLMs.** S3 and S4 rely on external LLMs. Figure 7 examines the impact of LLMs on emotion preference prediction. We observe that Qwen3-14B (Yang et al., 2025) performs better under the S3 strategy, while Qwen2.5-7B performs better under the S4 strategy. These results reveal an interesting phenomenon: *different LLMs exhibit varying preferences for different prompting strategies*. To further investigate, we conduct a statistical analysis of the two LLMs using their respective best strategies. Experimental results demonstrate that Qwen2.5-7B generally outperforms Qwen3-14B. These findings suggest that a larger LLM does not necessarily guarantee better alignment with human preferences. In this paper, we adopt Qwen2.5-7B as the default external LLM.

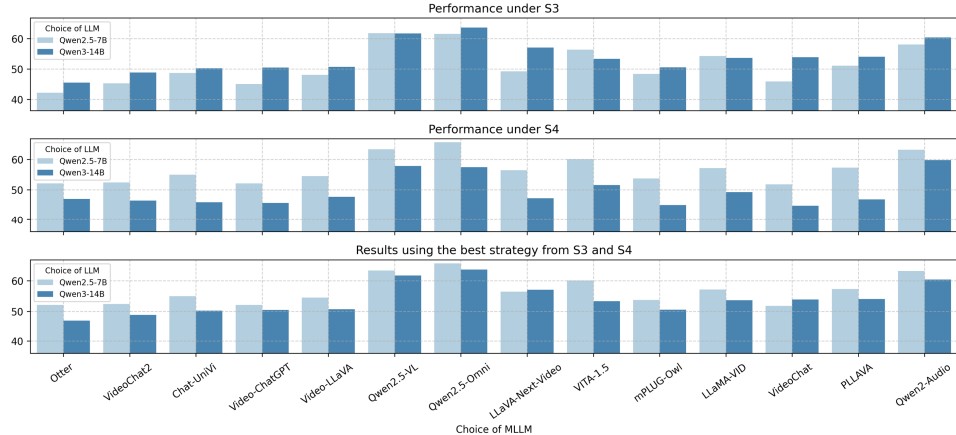

Figure 7: **Role of external LLMs.** From top to bottom, we show the performance of two LLMs under three configurations: *the S3 strategy*, *the S4 strategy*, and *the best of the S3 and S4 strategies*.

**Performance on Different MLLMs.** Table 2 presents the performance of MLLMs under their best strategies. Full results for four strategies are provided in Appendix G. Experimental results demonstrate that open-source models do not show a noticeable performance gap compared to closed-source models; some open-source models even surpass closed-source ones. Among all models, Qwen2.5-Omni achieves promising results, attaining 67.21% on the two-class WAF and 79.09% on swap consistency, highlighting its strong capability in multimodal information processing and human emotion understanding. Additionally, most models outperform random guessing (i.e., 50%), yet even the best-performing one falls short of the upper bound. This indicates the challenge of emotion preference decoding and suggests that further exploration is needed in the future.

Table 2: **Main results.** The second column indicates the optimal strategy for each MLLM.

| Model | S. | Rec. (2-Class) | | Rec. (3-Class) | | Swap |
|---|---|---|---|---|---|---|
| | | WAF(↑) | ACC(↑) | WAF(↑) | ACC(↑) | Cons. (↑) |
| *Open-source MLLMs* | | | | | | |
| VideoChat (Li et al., 2025b) | S4 | 51.79±1.21 | 52.31±0.98 | 44.88±1.54 | 40.77±1.57 | 40.85±1.83 |
| Video-ChatGPT (Maaz et al., 2024) | S4 | 52.10±0.94 | 52.13±0.98 | 43.97±0.44 | 40.42±0.35 | 45.91±1.13 |
| Otter (Li et al., 2025a) | S4 | 52.12±1.10 | 52.49±0.98 | 43.41±1.67 | 38.15±1.74 | 46.08±0.96 |
| VideoChat2 (Li et al., 2024b) | S4 | 52.38±0.10 | 52.49±0.09 | 45.88±0.96 | 43.82±0.44 | 41.46±1.74 |
| mPLUG-Owl (Ye et al., 2023) | S4 | 53.76±0.17 | 53.82±0.18 | 46.90±0.24 | 44.34±0.44 | 42.33±1.39 |
| Video-LLaVA (Lin et al., 2024) | S4 | 54.53±0.81 | 54.62±0.80 | 43.61±0.82 | 38.94±0.44 | 42.86±3.31 |
| Chat-UniVi (Jin et al., 2024) | S4 | 54.97±1.64 | 55.06±1.60 | 47.03±2.32 | 44.69±2.00 | 43.90±0.52 |
| LLaVA-Next-Video (Li et al., 2024a) | S4 | 56.41±0.98 | 56.57±0.98 | 52.84±1.25 | 50.78±1.31 | 53.92±2.00 |
| LLaMA-VID (Li et al., 2024c) | S4 | 57.10±0.63 | 57.10±0.62 | 50.42±1.17 | 47.13±1.66 | 45.12±0.70 |
| PLLAVA (Xu et al., 2024) | S4 | 57.29±0.18 | 57.55±0.18 | 54.02±0.05 | 52.61±0.00 | 53.83±1.22 |
| VITA-1.5 (Fu et al., 2025) | S4 | 60.08±0.61 | 60.12±0.62 | 57.08±0.16 | 56.01±0.09 | 59.06±1.22 |
| Qwen2-Audio (Chu et al., 2024) | S4 | 63.17±0.19 | 63.23±0.18 | 60.15±0.76 | 59.32±0.78 | 61.50±0.17 |
| Qwen2.5-VL (Bai et al., 2025) | S1 | 64.43±0.87 | 65.28±0.80 | 62.60±0.84 | 64.02±0.78 | 62.02±0.35 |
| Qwen2.5-Omni (Xu et al., 2025) | S2 | **67.21**±0.00 | **67.32**±0.00 | **65.29**±0.00 | **66.03**±0.00 | 79.09±0.00 |
| *Closed-source MLLMs* | | | | | | |
| GPT-4o (OpenAI, 2024) | S1 | 59.28±0.08 | 59.41±0.09 | 56.57±0.04 | 53.75±0.09 | 64.55±0.96 |
| Gemini-2.0-Flash (Google, 2025a) | S1 | 59.80±0.88 | 60.39±0.89 | 58.14±0.65 | 55.66±0.61 | 57.14±2.26 |
| Gemini-2.5-Flash (Google, 2025b) | S1 | 60.60±0.64 | 61.19±0.62 | 58.77±0.55 | 57.93±0.78 | 64.98±1.39 |
| GPT-4.1 (Achiam et al., 2023) | S1 | 60.75±0.18 | 60.75±0.18 | 59.23±0.35 | 59.67±0.26 | **80.84**±0.52 |
| Gemini-1.5-Pro (Team et al., 2024) | S1 | 60.79±0.68 | 61.55±0.62 | 59.06±0.66 | 60.37±0.61 | 62.37±0.35 |
| Gemini-1.5-Flash (Team et al., 2024) | S1 | 64.64±0.12 | 65.19±0.18 | 62.55±0.03 | 63.50±0.09 | 72.04±0.44 |

**Metric Correlation.** Figure 8a calculates the Pearson correlation coefficients (PCC) using scores from all strategies (see Appendix G), whereas Figure 8b uses scores from the optimal strategies (see Table 2). In Figure 8a, we observe a strong correlation between two-class and three-class *recognition performance*, whereas these metrics show only a relatively weak correlation with *swap consistency*. These results suggest that the two metrics capture distinct aspects of model capability: one reflects preference prediction accuracy, while the other reflects model robustness. A good model should perform well on both metrics. In Figure

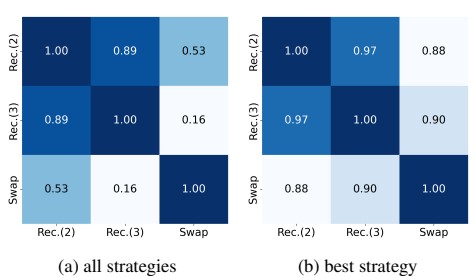

(a) all strategies      (b) best strategy

Figure 8: Metric correlation analysis.

8b, the correlation between these metrics is higher than in Figure 8a. This indicates that models with poor *recognition performance* may exhibit more varied *swap consistency*. But for well-performing models, these metrics remain related.

**Effect of Normal-Swapped Combination.** Table 2 reveals that swapping input orders may lead to distinct preference prediction results. This observation naturally raises the hypothesis: *Can we enhance the reliability of the results by aggregating the outputs of normal and swapped orders?* To verify this, we execute each model twice in normal order and twice in swapped order, generating four outcomes. We then apply majority voting to determine the final prediction. Figure 9 demonstrates the impact of this strategy across different MLLMs. We observe that most MLLMs benefit from this

approach. However, some exceptions exist, such as Chat-UniVi (Jin et al., 2024), indicating that the effectiveness of this strategy is model-dependent.

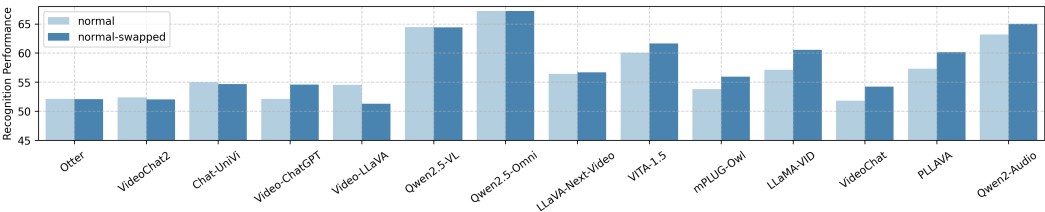

Figure 9: Normal-only vs. Normal-swapped combinations.

**Model-Based Crowdsourcing.** This section investigates the impact of model-based crowdsourcing. Specifically, we rank models based on their recognition performance and aggregate the predictions of the top-$k$ models. As shown in Figure 10, crowdsourcing generally improves performance, but the extent of the gain depends on $k$: a large $k$ introduces noise, whereas a small $k$ limits effectiveness. We then conduct an analysis from two perspectives: *(1) Necessity of normal-swapped combinations.* We examine the effectiveness of normal-swapped combinations in crowdsourcing. Unlike the findings for single models (see Figure 9), normal-swapped combinations yield only marginal improvements and may even degrade performance in some cases (e.g., Figures 10a and 10d). Therefore, we do not use normal-swapped combinations by default in crowdsourcing. *(2) Restricting the model scope.* Figures 10a~10c show performance when restricting the model scope, with the scope ranging from *all*, *open-source*, to *closed-source* models, respectively. Compared to no restrictions, restricting the model scope to closed-source models leads to a noticeable performance degradation, whereas limiting it to open-source models still yields promising results. This suggests that restricting evaluations to open-source models could reduce costs without significant performance loss.

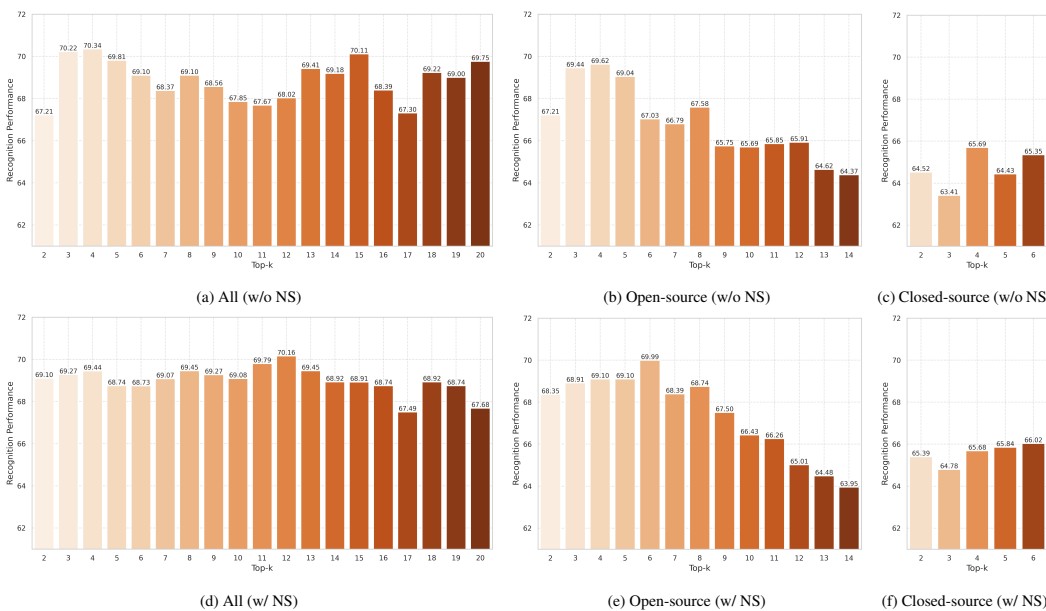

Figure 10: **Impact of model-based crowdsourcing on *recognition performance*.** In these figures, "w/o NS" and "w/ NS" denote whether normal-swapped combinations are used, while "all", "open-source", and "closed-source" indicate whether the model scope is restricted.

Besides its impact on *recognition performance*, Figure 11 further reveals its effect on *swap consistency*. We observe that selecting an appropriate $k$ also improves *swap consistency*, indicating greater robustness to input order changes with crowdsourcing. Interestingly, the optimal $k$ differs for *swap*

*consistency* and *recognition performance* (e.g., Figures 10a and 11a). Since a good model should perform well on both metrics, we should consider both when selecting $k$. For example, $k = 3$ or $4$ is a good choice when restricting the model scope to open-source models.

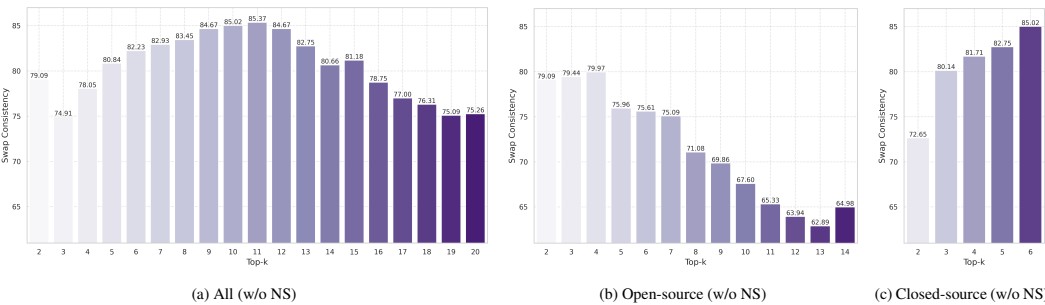

Figure 11: Impact of model-based crowdsourcing on *swap consistency*.

# 5 PRACTICAL APPLICATION

Figure 12(a) integrates preference prediction with the Bradley-Terry algorithm (Hunter, 2004) for model ranking. Specifically, suppose there are $M$ models and $N$ samples, where each sample is denoted as $x^n$ and each model as $m_i$. Taking the comparison between models $m_i$ and $m_j$ as an example, we first obtain their emotion descriptions for $x^n$, yielding $d_i^n$ and $d_j^n$. We then use judge MLLMs to predict preference. Inspired by the results of EmoPrefer-Bench, we use model-based crowdsourcing, restrict the model scope to open-source models, and set top-$k$=3. The predicted preference is denoted as $p_{ij}^n$. By repeating this process across multiple samples, we obtain the numbers of wins, losses, and ties between $m_i$ and $m_j$, denoted as $\text{win}_{ij}$, $\text{lose}_{ij}$, and $\text{tie}_{ij}$, respectively. These scores are stored in the matrix $\mathbf{W}$, and then the Bradley-Terry algorithm is used to estimate the advantages. Appendix F provides the detailed calculation process of the ranking algorithm.

Figure 12 provides a concrete example. Specifically, MER2025 (Lian et al., 2025b) is a prominent challenge in affective computing, featuring a track focused on descriptive emotions. With permission from the challenge organizers, we use participants' submissions for model ranking. Figure 12(b) visualizes the matrix W, and Figure 12(c) presents the estimated advantage scores.

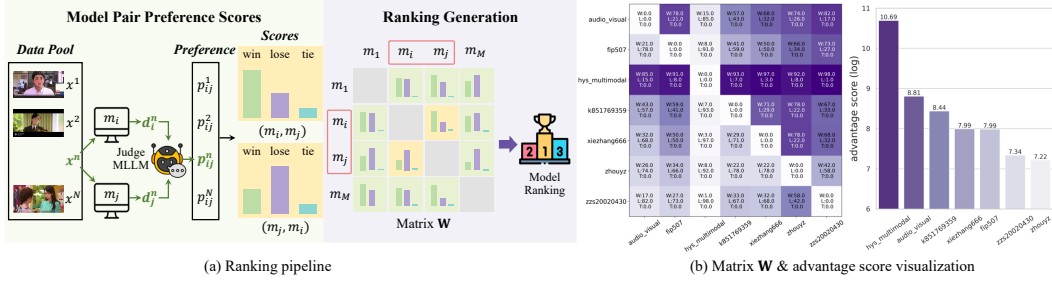

Figure 12: **Practical application.** (a) Ranking pipeline. We first employ pairwise models $m_i$ and $m_j$ to generate descriptions for $x^n$, obtaining $d_i^n$ and $d_j^n$. Next, we use MLLM-based judges to derive the preference prediction $p_{ij}^n$. By repeating this process, we collect the counts of wins, losses, and ties for each model pair comparison. These results are then used to compute the model rankings. (b) We visualize the matrix $\mathbf{W}$ and advantage scores derived from a concrete example.

# 6 CONCLUSION

This paper proposes *EmoPrefer*, a pioneering work that explores the capabilities of MLLMs in recognizing human emotion preferences. To this end, we introduce *EmoPrefer-Data* and establish

*EmoPrefer-Bench*, the first dataset and benchmark focused on human emotion preference. Extensive experiments show that different MLLMs favor different prompting strategies, and model-based crowdsourcing can enhance both recognition performance and model robustness. Meanwhile, we observe that current MLLMs still struggle with accurate emotion preference prediction. This highlights both the limitations of current MLLMs and the inherent challenges in emotion understanding.

**Limitations and Future Work.** This paper reports the zero-shot performance of MLLMs, without exploring more effective architectures or training paradigms. Such exploration is left for our future work. Beyond evaluation, we will leverage our dataset and the insights drawn from our benchmark to train reward models and apply reinforcement learning to maximize these rewards. This approach will enhance MLLMs' understanding of human emotions and, in turn, improve their emotional intelligence. Additionally, this paper focuses on emotion preference in descriptive emotion. Beyond descriptive emotions, other forms of emotion representation exist, including categorical emotions (e.g., using discrete emotion words) and dimensional emotions (e.g., using float-point values). In the future, we will extend our preference task to other emotion representations.

## ETHICS STATEMENT

We do not collect new data but instead use the raw samples from MER2024 with the permission of the owners. To construct EmoPrefer-Data, we hire multiple annotators. They are generously compensated for their work. During the annotation process, we provide a clear task description and utilize a well-designed annotation platform. Furthermore, we restrict the use of EmoPrefer-Data to non-commercial purposes under the CC BY-NC 4.0 license, which explicitly outlines the appropriate and responsible use of our dataset. As a result, no ethical concerns are raised in this paper.

## REPRODUCIBILITY STATEMENT

Our data and code are released at https://github.com/zeroQiaoba/AffectGPT/tree/master/EmoPrefer. For dataset annotation, we provide a detailed introduction in Appendix C. In summary, we have made every effort to ensure the reproducibility of this paper.

## ACKNOWLEDGEMENTS

This work is supported by the National Natural Science Foundation of China (No.62201572, No.62088101, No.62322120, No.61831022, No.62276259, No.U21B2010), the European Union's Horizon Europe Research and Innovation Programme under the Marie Skłodowska-Curie Actions grant agreement No.101126602 (Data4Healthcare), the Innovative Team Project of Multimodal Artificial Intelligence and Interaction in Guangdong Province (2025KCXTD040), the University of Oulu & The Academy of Finland Profi 7 Hybrid Intelligence (grant 352788), and Research Fellow project (grant 371019). We also wish to acknowledge the CSC – IT Center for Science, Finland, for computational resources.

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

APPENDIX

## A    LLM USAGE

In this paper, LLMs are merely used to polish language, such as improving the clarity, coherence, and fluency. We do not leverage LLMs in the generation of research ideas. Our research ideation involves deep domain expertise, creativity, and an understanding of unresolved scientific challenges, all of which rely on our insights into the field of affective computing. Therefore, LLMs serve as supportive tools for language enhancement rather than as drivers of idea breakthroughs.

## B    RELATED WORKS

### B.1    DESCRIPTIVE EMOTION REPRESENTATION

Emotion is intrinsically linked to human cognition, decision-making, and behavior (Cacioppo & Gardner, 1999; Lerner et al., 2015; Dolan, 2002) and plays a pivotal role in artificial intelligence (Minsky, 1986). Emotion representation methods serve as the foundation, aiming to map complex human emotions into computable values (Gunes et al., 2011). Within the field of emotion representation, two predominant paradigms exist: categorical representation and dimensional representation.

**Categorical Representation.**    Rooted in psychological theory, researchers classify human emotions into discrete categories. For instance, Ekman & Keltner (1970) proposed that a set of universal emotional states, termed basic emotions, exists across all human cultures, including *anger*, *disgust*, *fear*, *happiness*, *sadness*, and *surprise*. Later, Plutchik (1980) introduced eight primary emotions, organized in opposing pairs and arranged by intensity, which can combine to form secondary emotions. However, human emotions are far more complex than these six or eight labels (Lindquist & Barrett, 2008). Restricting the emotional space to such limited categories inevitably overlooks nuanced emotional experiences. To address this, Lian et al. (2025c) proposed recognizing emotions in an open-vocabulary manner, enabling the prediction of arbitrary emotion categories. This approach extends beyond traditional categorical representation paradigms.

**Dimensional Representation.**    Rather than discrete categories, dimensional representation theory models human emotions as points in a continuous multi-dimensional space. For example, Russell & Mehrabian (1977) proposed the Valence–Arousal–Dominance (VAD) model, where valence reflects the positivity of an emotion, arousal reflects its level of excitement, and dominance reflects the perceived sense of control in a situation. Later, Russell (1980) introduced a simpler Circumplex Model of Affect, reducing emotions to a two-dimensional space—valence and arousal. However, dimensional emotions are abstract and less intuitive than discrete labels (e.g., happy), limiting their practical use in downstream tasks where simpler categorical models are preferred.

**Descriptive Representation.**    Neither categorical nor dimensional representations fully capture human cognition in emotion perception and understanding, rendering the emotion understanding process a black-box operation. To address this limitation, Lian et al. (2023; 2025a) proposed a novel representation strategy called *descriptive emotions*. They leverage free-form natural language to visualize the human emotion understanding process by incorporating multimodal clues and additional analysis. Within its internal analysis, this representation further accounts for emotions' temporal dynamics, intensity, and uncertainty, thereby providing more human-like emotion representations. Despite its advantages, evaluating the quality of such open-ended descriptions remains a non-trivial challenge. Previous studies rely on costly human annotators to identify human-preferred descriptions. In this paper, we propose *EmoPrefer*, which leverages MLLMs to achieve more cost-efficient preference annotation, offering a meaningful complement to existing evaluation strategies for descriptive emotions. To the best of our knowledge, this is the first work to explore the capabilities of MLLMs in emotion preference decoding.

### B.2    LLM-BASED JUDGE

LLM-as-a-Judge aims to leverage LLMs as evaluators to replace traditional human-driven assessments, offering a cost-effective evaluation solution (Gu et al.). Beyond evaluation, these LLM-based judges can also serve as reward models in reinforcement learning (Christiano et al., 2017) or act as verifiers to select the best-of-N responses from multiple candidates (Cobbe et al., 2021). To assess

the effectiveness of LLM-based judges, it is necessary to measure the agreement between LLM-generated responses and human judgments. This necessitates a series of benchmark datasets with human annotations. For example, MT-Bench (Zheng et al., 2023) consists of multi-turn questions designed to evaluate a chatbot's conversational abilities and adherence to instructions. FairEval (Wang et al., 2024a) provides human-annotated preferences for responses generated by ChatGPT and Vicuna. JudgeBench (Tan et al., 2025) evaluates LLM-based judges on challenging response pairs requiring advanced reasoning skills, such as knowledge, math, and coding. These benchmarks primarily focus on text-only preference tasks. In contrast, MLLM-as-a-Judge (Chen et al., 2024) extends such tasks to single-image or image-sequence preference evaluations, broadening the range of involved modalities. In this paper, we propose *EmoPrefer-Data*, the first multimodal preference dataset specifically designed for human emotions. Unlike previous benchmarks that focus on text-only, single-image, or image-sequence inputs, human emotions are often conveyed through subtle behaviors embedded in multimodal cues. To achieve this, EmoPrefer-Data not only expands the supported modalities to full multimodal scenarios (including audio, text, and video) but also serves as the first dataset centered on human emotions. Therefore, our work provides a valuable data resource for current research on LLM-based judges.

## C   DETAILS OF EMOPREFER-DATA

To construct EmoPrefer-Data, we provide pairwise emotion descriptions for each video and then recruit multiple annotators to evaluate which description better aligns with the character's emotional state. This section provides further details about the construction and characteristics of our dataset.

### C.1   DATA VISUALIZATION

Figure 13 presents three examples from EmoPrefer-Data. The original videos capture real people's emotional expressions in uncontrolled settings. To preserve privacy, we apply a style-transfer technique to anonymize the individuals. The use of this dataset has been authorized by its owner. As shown in Figure 13, these videos primarily depict a single front-facing individual and include complete audio recordings, ensuring the full expression of emotions for the target speaker.

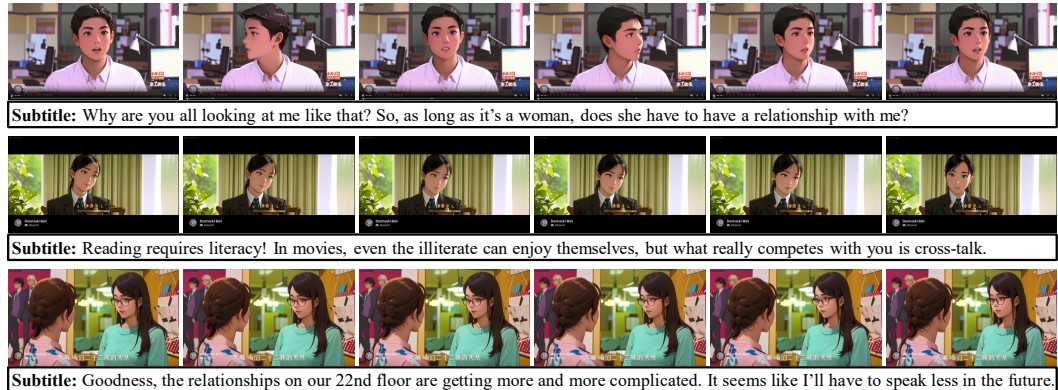

**Subtitle:** Why are you all looking at me like that? So, as long as it's a woman, does she have to have a relationship with me?

**Subtitle:** Reading requires literacy! In movies, even the illiterate can enjoy themselves, but what really competes with you is cross-talk.

**Subtitle:** Goodness, the relationships on our 22nd floor are getting more and more complicated. It seems like I'll have to speak less in the future.

Figure 13: **Data visualization.** We cartoonize these video data to eliminate privacy concerns.

### C.2   DESCRIPTION STATISTICS

To ensure the quality of description pairs, we leverage data in two benchmark datasets: MERR-Fine and MER-Caption+. To visualize their description richness, we extract emotion words and nouns from these descriptions using Qwen2.5-7B with the following prompts:

**Emotion Words:** *Please assume the role of an expert in the field of emotions. We provide clues that may be related to the emotions of the characters. Based on the provided clues, please identify the emotional states of the main characters. Please separate different emotional categories with*

*commas and output only the clearly identifiable emotional categories in a list format. If none are identified, please output an empty list.*

**Nouns:** *We provide clues that may be related to the emotions of the characters. Please extract all nouns from the provided clues. Please separate different words with commas and output in a list format. If none are identified, please output an empty list.*

### C.3 ANNOTATION PLATFORM LAYOUT

Figure 14 shows the layout of the annotation platform. During the annotation process, we provide two descriptions for each video. Annotators should select the preferred description, i.e., the one that better matches the character's emotional state, based on the video content. Annotators can choose from three options: (1) *description 1 is better*; (2) *description 2 is better*; (3) *tie*.

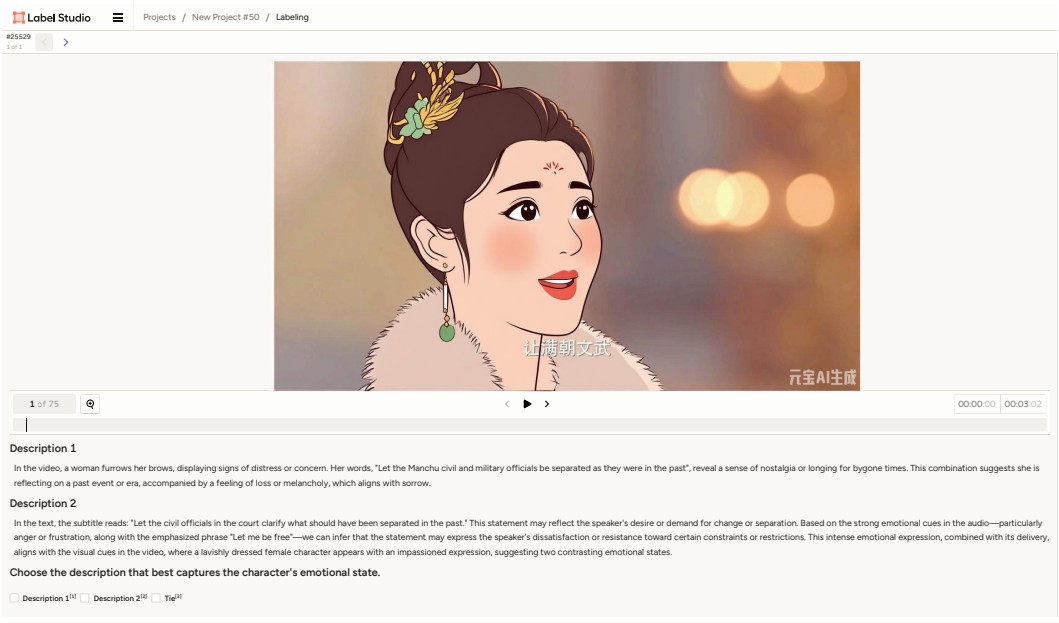

Figure 14: **Annotation platform layout.** We provide pairwise descriptions for each video and ask annotators to select the one that better reflects the character's emotional state. In this figure, we apply cartoonization to the video data to eliminate privacy concerns.

## D DETAILS OF MLLM-BASED STRATEGIES

Figures 15∼17 illustrate the prompts for S1∼S4. As shown in these figures, the length of the reasoning chains increases progressively from S1 to S4.

### D.1 DETAILS OF S1

For S1, we provide the MLLM with a video and two descriptions, instructing it to identify the better one. Figure 15 shows the prompt used in S1.

> We provide two descriptions for a given input. Please determine which one is better aligned with the input content. If both descriptions are equally aligned with the input content, please output 'tie'. Directly output the answer without additional reasoning.
> Video: {Video}
> D1: {Description_1}
> D2: {Description_2}

Figure 15: Prompt for S1.

## D.2 DETAILS OF S2 AND S3

S2 and S3 decompose S1 into two stages: description generation and preference prediction. Figure 16 illustrates the prompts for S2 and S3.

> Please provide a detailed description of a given video, with a particular focus on the emotions portrayed.
> Video: {Video}
> ------------------------------------------------------------------------------------------------------------
> We provide a true description and two predicted descriptions. Please determine which one is better aligned with the true description. If both descriptions are equally aligned with the true description, please output 'tie'. Directly output the answer without additional reasoning.
> Ground Truth: {Output of the First Step}
> D1: {Description_1}
> D2: {Description_2}

Figure 16: Prompt for S2 and S3.

## D.3 DETAILS OF S4

S4 decomposes S1 into three steps: description generation, preference reasoning, and answer extraction. Figure 17 provides details of the prompt used in S4.

> Please provide a detailed description of a given video, with a particular focus on the emotions portrayed.
> Video: {Video}
> ------------------------------------------------------------------------------------------------------------
> We provide a true description and two predicted descriptions. Please determine which one is better aligned with the true description. If both descriptions are equally aligned with the true description, please output 'tie'. Please output the answer along with the reasoning process.
> Ground Truth: {Output of the First Step}
> D1: {Description_1}
> D2: {Description_2}
> ------------------------------------------------------------------------------------------------------------
> Based on the provided description, please determine which one is better aligned with the ground truth description. The output should be 'D1', 'D2', or 'tie'.
> Description: {Output of the Second Step}

Figure 17: Prompt for S4.

## E    PROMPT TEMPLATE FOR SWAP CONSISTENCY

Figure 18 visualizes the prompts for normal and swapped inputs. We require the model to be robust to input order changes.

| |
|---|
| We provide two descriptions for a given input. Please determine which one is better aligned with the input content. If both descriptions are equally aligned with the input content, please output 'tie'. Directly output the answer without additional reasoning. 
 Video: {Video} 
 D1: {Description_1} 
 D2: {Description_2} 
 **Output:** {D1} |
| We provide two descriptions for a given input. Please determine which one is better aligned with the input content. If both descriptions are equally aligned with the input content, please output 'tie'. Directly output the answer without additional reasoning. 
 Video: {Video} 
 D1: {Description_2} 
 D2: {Description_1} 
 **Output:** {D2} |

Figure 18: **Swap consistency.** This figure illustrates an ideal case, where the model prefers consistent descriptions regardless of input order.

## F    DETAILS OF BRADLEY-TERRY ALGORITHM

Suppose there are $M$ models, each denoted as $m_i$. In this algorithm, each model $m_i$ is associated with a positive parameter $\theta_i$, which quantifies its relative advantage. The probabilities that model $m_i$ wins, loses, or ties against model $m_j$ are defined as follows:

$$P(m_i > m_j) = \frac{\theta_i}{\theta_i + \theta_j + \beta}, \; P(m_i < m_j) = \frac{\theta_j}{\theta_i + \theta_j + \beta}, \; P(m_i \sim m_j) = \frac{\beta}{\theta_i + \theta_j + \beta}, \quad (2)$$

where $\beta$ controls the likelihood of ties. The parameters $\theta$ can then be estimated by maximizing the following likelihood function. Models with higher $\theta$ values are ranked higher.

$$\max \mathcal{L}(\theta) = \prod_{i<j} \left( \frac{\theta_i}{\theta_i + \theta_j + \beta} \right)^{\text{win}_{ij}} \left( \frac{\theta_j}{\theta_i + \theta_j + \beta} \right)^{\text{lose}_{ij}} \left( \frac{\beta}{\theta_i + \theta_j + \beta} \right)^{\text{tie}_{ij}}, \quad (3)$$

where $\text{win}_{ij}$, $\text{lose}_{ij}$, and $\text{tie}_{ij}$ denote the numbers of wins, losses, and ties between model $m_i$ and model $m_j$, respectively.

## G    MLLM PERFORMANCE ACROSS PROMPTING STRATEGIES

Table 3 presents the performance of MLLMs across four prompting techniques. Our analysis reveals that different MLLMs favor distinct prompting strategies. Notably, high-performing MLLMs on multimodal tasks (such as Qwen2.5-Omni and Qwen2.5-VL) favor S1 or S2, whereas other MLLMs generally achieve better performance with S3 or S4.

Table 3: Performance of MLLMs under four prompting strategies.

| Model | S. | Rec. (2-Class) | | Rec. (3-Class) | | Swap Cons. (↑) |
|---|---|---|---|---|---|---|
| | | WAF(↑) | ACC(↑) | WAF(↑) | ACC(↑) | |
| VideoChat (Li et al., 2025b) | S1 | 38.81±1.68 | 50.18±1.51 | 36.62±1.30 | 43.55±0.70 | 13.59±0.52 |
| | S2 | 38.45±0.76 | 49.82±0.27 | 33.90±0.74 | 38.07±0.78 | 14.90±2.00 |
| | S3 | 45.94±1.62 | 51.60±0.62 | 33.79±2.26 | 27.70±1.92 | 41.99±2.44 |
| | S4 | 51.79±1.21 | 52.31±0.98 | 44.88±1.54 | 40.77±1.57 | 40.85±1.83 |
| Video-ChatGPT (Maaz et al., 2024) | S1 | 31.98±0.08 | 48.67±0.18 | 19.81±1.03 | 18.29±0.87 | 46.69±0.17 |
| | S2 | 32.06±0.00 | 48.85±0.00 | 2.83±0.46 | 3.40±0.26 | 93.12±0.44 |
| | S3 | 45.04±0.03 | 47.87±0.27 | 37.99±0.38 | 35.71±0.52 | 45.30±0.00 |
| | S4 | 52.10±0.94 | 52.13±0.98 | 43.97±0.44 | 40.42±0.35 | 45.91±1.13 |
| Otter (Li et al., 2025a) | S1 | 32.06±0.00 | 48.85±0.00 | 0.07±0.00 | 1.92±0.00 | 100.00±0.00 |
| | S2 | 32.06±0.00 | 48.85±0.00 | 0.07±0.00 | 1.92±0.00 | 98.95±0.00 |
| | S3 | 42.21±0.13 | 46.45±0.09 | 34.51±0.10 | 31.79±0.09 | 46.95±0.44 |
| | S4 | 52.12±1.10 | 52.49±0.98 | 43.41±1.67 | 38.15±1.74 | 46.08±0.96 |
| VideoChat2 (Li et al., 2024b) | S1 | 39.79±0.53 | 42.63±0.18 | 37.01±0.87 | 37.63±0.87 | 36.24±0.17 |
| | S2 | 42.02±2.48 | 49.64±2.04 | 40.09±2.30 | 45.12±1.57 | 21.60±0.17 |
| | S3 | 45.31±1.64 | 48.31±1.60 | 37.73±1.62 | 34.84±1.05 | 38.85±1.74 |
| | S4 | 52.38±0.10 | 52.49±0.09 | 45.88±0.96 | 43.82±0.44 | 41.46±1.74 |
| mPLUG-Owl (Ye et al., 2023) | S1 | 39.13±2.32 | 49.20±1.07 | 33.32±3.08 | 33.62±2.79 | 22.13±2.61 |
| | S2 | 37.59±0.32 | 48.31±0.18 | 31.71±0.83 | 33.01±1.48 | 18.47±0.70 |
| | S3 | 48.41±0.24 | 51.33±0.36 | 42.87±0.40 | 40.16±0.44 | 42.25±5.31 |
| | S4 | 53.76±0.17 | 53.82±0.18 | 46.90±0.24 | 44.34±0.44 | 42.33±1.39 |
| Video-LLaVA (Lin et al., 2024) | S1 | 32.06±0.00 | 48.85±0.00 | 31.04±0.00 | 47.91±0.00 | 0.00±0.00 |
| | S2 | 32.06±0.00 | 48.85±0.00 | 31.04±0.00 | 47.91±0.00 | 0.09±0.09 |
| | S3 | 48.10±0.29 | 51.69±0.18 | 39.61±0.80 | 35.63±1.13 | 39.55±0.35 |
| | S4 | 54.53±0.81 | 54.62±0.80 | 43.61±0.82 | 38.94±0.44 | 42.86±3.31 |
| Chat-UniVi (Jin et al., 2024) | S1 | 36.66±0.26 | 48.49±0.18 | 35.52±0.28 | 47.47±0.09 | 10.71±1.13 |
| | S2 | 35.20±0.05 | 49.29±0.09 | 33.92±0.21 | 47.30±0.26 | 6.10±0.17 |
| | S3 | 48.66±0.14 | 51.15±0.18 | 44.05±0.02 | 42.25±0.09 | 44.34±2.53 |
| | S4 | 54.97±1.64 | 55.06±1.60 | 47.03±2.32 | 44.69±2.00 | 43.90±0.52 |
| LLaVA-Next-Video (Li et al., 2024a) | S1 | 35.13±0.00 | 50.27±0.00 | 34.04±0.00 | 49.30±0.00 | 2.79±0.00 |
| | S2 | 33.23±0.00 | 49.38±0.00 | 32.18±0.00 | 48.43±0.00 | 2.26±0.00 |
| | S3 | 49.25±0.03 | 51.42±0.09 | 44.93±0.23 | 42.16±0.17 | 51.74±0.52 |
| | S4 | 56.41±0.98 | 56.57±0.98 | 52.84±1.25 | 50.78±1.31 | 53.92±2.00 |
| LLaMA-VID (Li et al., 2024c) | S1 | 33.03±0.19 | 49.29±0.09 | 31.99±0.19 | 48.34±0.09 | 0.61±0.26 |
| | S2 | 32.25±0.20 | 48.93±0.09 | 31.23±0.19 | 48.00±0.09 | 0.09±0.09 |
| | S3 | 54.30±1.31 | 56.48±1.24 | 47.76±1.37 | 44.16±1.13 | 44.16±1.31 |
| | S4 | 57.10±0.63 | 57.10±0.62 | 50.42±1.17 | 47.13±1.66 | 45.12±0.70 |
| PLLAVA (Xu et al., 2024) | S1 | 35.79±0.00 | 50.44±0.00 | 34.69±0.00 | 49.48±0.00 | 5.05±0.00 |
| | S2 | 32.84±0.00 | 49.20±0.00 | 31.80±0.00 | 48.26±0.00 | 1.39±0.00 |
| | S3 | 51.13±0.44 | 53.02±0.44 | 47.76±0.01 | 46.43±0.09 | 51.22±0.87 |
| | S4 | 57.29±0.18 | 57.55±0.18 | 54.02±0.05 | 52.61±0.00 | 53.83±1.22 |
| VITA-1.5 (Fu et al., 2025) | S1 | 48.85±0.00 | 48.85±0.00 | 47.44±0.00 | 47.91±0.00 | 70.73±0.00 |
| | S2 | 56.34±0.00 | 60.04±0.00 | 54.74±0.00 | 58.89±0.00 | 41.99±0.00 |
| | S3 | 56.38±0.37 | 57.28±0.27 | 52.85±0.70 | 51.83±0.78 | 58.54±0.17 |
| | S4 | 60.08±0.61 | 60.12±0.62 | 57.08±0.16 | 56.01±0.09 | 59.06±1.22 |
| Qwen2-Audio (Chu et al., 2024) | S1 | 45.13±0.37 | 54.62±0.27 | 39.91±0.08 | 50.96±0.09 | 6.18±0.09 |
| | S2 | 50.52±0.24 | 54.44±0.44 | 44.18±0.04 | 48.17±0.44 | 14.81±0.17 |
| | S3 | 58.10±0.29 | 58.44±0.36 | 56.16±0.46 | 56.36±0.61 | 63.94±0.70 |
| | S4 | 63.17±0.19 | 63.23±0.18 | 60.15±0.76 | 59.32±0.78 | 61.50±0.17 |
| Qwen2.5-VL (Bai et al., 2025) | S1 | 64.43±0.87 | 65.28±0.80 | 62.60±0.84 | 64.02±0.78 | 62.02±0.35 |
| | S2 | 62.42±0.99 | 62.79±0.98 | 60.63±0.97 | 61.59±0.96 | 68.90±0.61 |
| | S3 | 61.79±1.41 | 62.61±1.33 | 58.38±1.62 | 56.53±1.48 | 58.71±1.05 |
| | S4 | 63.37±3.47 | 63.59±3.37 | 60.11±2.78 | 59.23±2.44 | 58.19±1.92 |
| Qwen2.5-Omni (Xu et al., 2025) | S1 | 63.36±0.00 | 63.41±0.00 | 61.54±0.00 | 62.20±0.00 | 73.87±0.00 |
| | S2 | 67.21±0.00 | 67.32±0.00 | 65.29±0.00 | 66.03±0.00 | 79.09±0.00 |
| | S3 | 61.54±0.07 | 61.72±0.09 | 59.05±0.06 | 56.62±0.00 | 72.65±0.52 |
| | S4 | 65.69±0.29 | 65.81±0.27 | 63.48±0.29 | 62.54±0.35 | 67.94±0.52 |

## H RELATIONSHIP ANALYSIS BETWEEN HUMANS AND MLLMS

We analyze the correlation between MLLMs and humans using the examples mentioned in Section 5. Specifically, we first recruit multiple human annotators to perform preference annotations. Then, we examine this correlation from two perspectives: *model-pair similarity* and *ranking similarity*.

### H.1 MODEL-PAIR SIMILARITY

Taking the comparison between $m_i$ and $m_j$ as an example. We denote the counts of wins, losses, and ties between $m_i$ and $m_j$ for human annotators as $\text{win}_{ij}$, $\text{loss}_{ij}$, and $\text{tie}_{ij}$, and those for MLLMs as $\hat{\text{win}}_{ij}$, $\hat{\text{loss}}_{ij}$, and $\hat{\text{tie}}_{ij}$. To evaluate the similarity between humans and MLLMs, we first convert these counts into three categories:

$$y_{ij} = \begin{cases} \text{win}, & \text{win}_{ij} > \text{loss}_{ij} \\ \text{lose}, & \text{win}_{ij} < \text{loss}_{ij} \\ \text{tie}, & \text{win}_{ij} = \text{loss}_{ij} \end{cases}, \ \hat{y}_{ij} = \begin{cases} \text{win}, & \hat{\text{win}}_{ij} > \hat{\text{loss}}_{ij} \\ \text{lose}, & \hat{\text{win}}_{ij} < \hat{\text{loss}}_{ij} \\ \text{tie}, & \hat{\text{win}}_{ij} = \hat{\text{loss}}_{ij} \end{cases}. \quad (4)$$

Figure 19 presents the model-pair comparison results for humans and MLLMs.

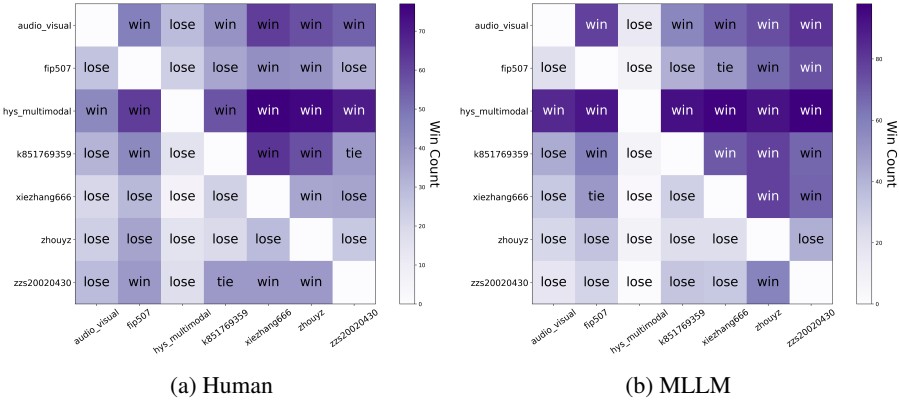

(a) Human                    (b) MLLM

Figure 19: Model-pair comparison results.

We then compute the similarity score between them:

$$\text{Similarity} = \frac{\sum_{i=1}^{M} \sum_{j=i+1}^{M} \mathbb{I}[y_{ij} = \hat{y}_{ij}]}{\sum_{i=1}^{M} \sum_{j=i+1}^{M} 1}, \quad (5)$$

where $\mathbb{I}(\cdot)$ is the indicator function. Experimental analysis reveals an 80.95% agreement rate, indicating strong alignment between humans and MLLMs.

### H.2 RANKING SIMILARITY

Using the Bradley-Terry algorithm (see Appendix F), we can derive model-level rankings from pairwise comparisons in Figure 19. In this section, we further evaluate the *ranking similarity* between MLLMs and humans. Specifically, we use Spearman's rank correlation coefficient $\rho$ to quantify this similarity. Let $\text{Rank}_A(\cdot)$ and $\text{Rank}_B(\cdot)$ be two ranking functions. For each model $m_i$, its rank in the two rankings is denoted as $\text{Rank}_A(m_i)$ and $\text{Rank}_B(m_i)$. Spearman's $\rho$ is computed as:

$$\rho = 1 - \frac{6 \sum_{i=1}^{M} [\text{Rank}_A(m_i) - \text{Rank}_B(m_i)]^2}{M(M^2 - 1)}. \quad (6)$$

This coefficient $\rho$ ranges from -1 to 1, with higher values indicating stronger positive correlation. Through experimental analysis, we observe that $\rho$ reaches 0.8571, a relatively high score, indicating good consistency between human and MLLM in decoding emotion preferences.

# I    RATIONALE FOR CHOOSING BINARY WAF AS THE DEFAULT METRIC

**Theoretical Basis.**   First, we opt for binary classification rather than tri-class because ties contain higher ambiguity than clear preferences (e.g., when one description is distinctly better than another), as we discussed in Section 2. Binary performance focuses on more definitively labeled comparisons, enabling more reliable conclusions.  Second, we chose WAF over ACC due to the inherent class imbalance in EmoPrefer-Data. As shown in Table 4, the $d_1$ category slightly outnumbers the $d_2$ category.  In imbalanced datasets, ACC is dominated by the majority class, whereas WAF better accounts for class imbalance. Therefore, we adopted WAF as our primary metric. Combining these two considerations, we use binary WAF as the default evaluation metric.

Table 4: **Class distribution in EmoPrefer-Data.** For a given video and two descriptions $(x, d_1, d_2)$, the preference label falls into one of three categories: $d_1$, $d_2$, or tie.

| Preference | Percentage |
|:---:|:---:|
| $d_1$ | 50.17% |
| $d_2$ | 47.91% |
| tie | 1.92% |

**Experimental Analysis.**   Using the scores from Table 2, we analyze the correlation between different metrics.  As shown in Figure 20, PCC scores between these metrics are relatively high, indicating strong inter-metric correlations.  Meanwhile, we examine the ranking consistency across metrics. Although the rankings from different metrics are not identical (see Table 2), their overall trends are similar. For instance, Qwen2.5-Omni consistently performs best in both binary and tri-class metrics, whereas VideoChat ranks relatively poorly across all metrics.

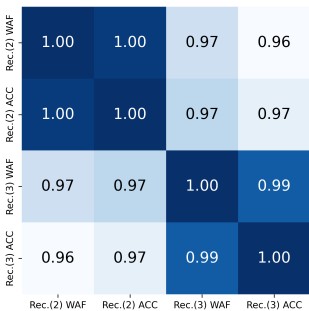

Figure 20: PCC scores between different metrics.

# J    INFERENCE COST COMPARISON

In this section, we fix the MLLM to Qwen2.5-Omni and report the inference costs across S1∼S4. As shown in Table 5, S1 is the fastest strategy due to its fewer inference steps. Meanwhile, S3/S4 runs faster than S2. The primary reason is that we leverage vLLM (Kwon et al., 2023) to accelerate the inference process of external LLMs in S3/S4, resulting in reduced inference time.

Table 5: Inference cost analysis for different prompting strategies.

| Promtping Strategy | Inference Time per Sample (seconds) |
|:---:|:---:|
| S1 | 1.3875 |
| S2 | 5.4188 |
| S3 | 4.0442 |
| S4 | 4.9370 |

## K VISUALIZATION OF INTERNAL OUTPUTS FOR S1∼S4

We employ Video-ChatGPT as the MLLM and visualize its outputs under S1∼S4 strategies.

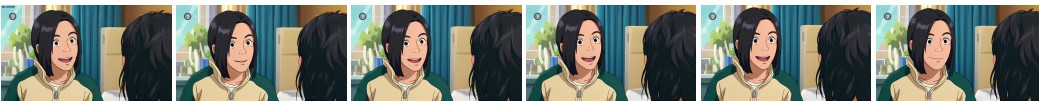

**Description_1:** In the video, a young woman displays a facial expression characterized by widened eyes and arched lips, indicating surprise, shock, or even anger. Her negative tone and intonation further support this inference.
**Description_2:** In the text, caption content: "Our unfinished business." This sentence might be someone's response to a particular situation or dialogue. Based on the video clues showing the woman's happiness, excitement, or curiosity, and her body language indicating of comfort and ease, we can infer that this sentence might carry a casual or playful tone. However, based on the audio clues describing tension, urgency, and pressure, and the use of the phrase "our business" indicating responsibility or obligation, we can also infer that this sentence might carry a sense of frustration, anger, or anxiety. Therefore, this sentence may be a sarcastic or provocative response, expressing the complex emotional state of the person in the situation.

***Strategy S1***
We provide two descriptions for a given input. Please determine which one is better aligned with the input content. If both descriptions are equally aligned with the input content, please output 'tie'. Directly output the answer without additional reasoning.
Video: {Video}
D1: {Description_1}
D2: {Description_2}
**Output:** Both descriptions are equally aligned with the input content, as they both describe the same scene with a young woman displaying a facial expression characterized by widened eyes and arched lips, indicating surprise, shock, or anger. The audio clues in the video suggest tension, urgency, and pressure, while the caption content "Our unfinished business" implies a casual or playful tone. The woman's body language indicates comfort and ease, which further supports the idea that the sentence might carry a casual or playful tone. However, the audio clues also suggest that the situation could be tense or urgent, which might lead to a sarcastic or provocative response. Overall, both descriptions provide plausible interpretations of the scene, making it difficult to determine which one is more accurate.

***Strategy S2***
Please provide a detailed description of a given video, with a particular focus on the emotions portrayed.
Video: {Video}
**Output:** The video shows a young girl talking to a young woman, and the girl is crying. The young woman appears to be concerned and tries to console the girl. The girl is sitting on a couch and the young woman is standing in front of her. The emotions displayed in the video are of sadness and concern.
--------------------------------------------------------------------------------------------------------------------------
We provide a true description and two predicted descriptions. Please determine which one is better aligned with the true description. If both descriptions are equally aligned with the true description, please output 'tie'. Directly output the answer without additional reasoning.
Ground Truth: {Output of the First Step}
D1: {Description_1}
D2: {Description_2}
**Output:** In the video, a young woman displays a facial expression characterized by widened eyes and arched lips, indicating surprise, shock, or even anger. Her negative tone and intonation further support this inference. The young girl is sitting on a couch, and the young woman is standing in front of her. The emotions displayed in the video are of sadness and concern. The young woman is trying to console the girl, which suggests a sense of empathy and care. The presence of a couch and the fact that the girl is sitting on it imply a casual and comfortable setting. The young woman's facial expression and body language indicate that she is trying to help the girl feel better, which further supports the idea that the situation is not serious or urgent.

Figure 21: Visualization of the internal outputs of S1 and S2.

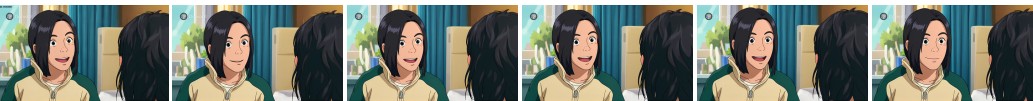

**Description_1:** In the video, a young woman displays a facial expression characterized by widened eyes and arched lips, indicating surprise, shock, or even anger. Her negative tone and intonation further support this inference.

**Description_2:** In the text, caption content: "Our unfinished business." This sentence might be someone's response to a particular situation or dialogue. Based on the video clues showing the woman's happiness, excitement, or curiosity, and her body language indicating of comfort and ease, we can infer that this sentence might carry a casual or playful tone. However, based on the audio clues describing tension, urgency, and pressure, and the use of the phrase "our business" indicating responsibility or obligation, we can also infer that this sentence might carry a sense of frustration, anger, or anxiety. Therefore, this sentence may be a sarcastic or provocative response, expressing the complex emotional state of the person in the situation.

*Strategy S3*

Please provide a detailed description of a given video, with a particular focus on the emotions portrayed.
Video: {Video}
**Output:** The video shows a young girl talking to a young woman, and the girl is crying. The young woman appears to be concerned and tries to console the girl. The girl is sitting on a couch and the young woman is standing in front of her. The emotions displayed in the video are of sadness and concern.

------------------------------------------------------------------------------------------------------------------------------

We provide a true description and two predicted descriptions. Please determine which one is better aligned with the true description. If both descriptions are equally aligned with the true description, please output 'tie'. Directly output the answer without additional reasoning.
Ground Truth: {Output of the First Step}
D1: {Description_1}
D2: {Description_2}
**Output:** D1

Figure 22: Visualization of the internal outputs of S3.

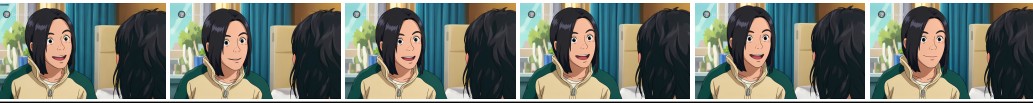

**Description_1:** In the video, a young woman displays a facial expression characterized by widened eyes and arched lips, indicating surprise, shock, or even anger. Her negative tone and intonation further support this inference.

**Description_2:** In the text, caption content: "Our unfinished business." This sentence might be someone's response to a particular situation or dialogue. Based on the video clues showing the woman's happiness, excitement, or curiosity, and her body language indicating of comfort and ease, we can infer that this sentence might carry a casual or playful tone. However, based on the audio clues describing tension, urgency, and pressure, and the use of the phrase "our business" indicating responsibility or obligation, we can also infer that this sentence might carry a sense of frustration, anger, or anxiety. Therefore, this sentence may be a sarcastic or provocative response, expressing the complex emotional state of the person in the situation.

*Strategy S4*

Please provide a detailed description of a given video, with a particular focus on the emotions portrayed.

Video: {Video}

**Output:** The video shows a young girl talking to a young woman, and the girl is crying. The young woman appears to be concerned and tries to console the girl. The girl is sitting on a couch and the young woman is standing in front of her. The emotions displayed in the video are of sadness and concern.

---------------------------------------------------------------------------------------------------------------------------

We provide a true description and two predicted descriptions. Please determine which one is better aligned with the true description. If both descriptions are equally aligned with the true description, please output 'tie'. Please output the answer along with the reasoning process.

Ground Truth: {Output of the First Step}

D1: {Description_1}

D2: {Description_2}

**Output:** Based on the provided ground truth description and the two predicted descriptions, let's analyze each one:### Ground Truth Description:- A young girl is crying.- A young woman is concerned and trying to console the girl.- The girl is sitting on a couch.- The young woman is standing in front of her.- The emotions displayed are sadness and concern.### Predicted Descriptions:1. **D1**: "In the video, a young woman displays a facial expression characterized by widened eyes and arched lips, indicating surprise, shock, or even anger. Her negative tone and intonation further support this inference."2. **D2**: "In the text, caption content: 'Our unfinished business.' This sentence might be someone's response to a particular situation or dialogue. Based on the video clues showing the woman's happiness, excitement, or curiosity, and her body language indicating of comfort and ease, we can infer that this sentence might carry a casual or playful tone. However, based on the audio clues describing tension, urgency, and pressure, and the use of the phrase 'our business' indicating responsibility or obligation, we can also infer that this sentence might carry a sense of frustration, anger, or anxiety. Therefore, this sentence may be a sarcastic or provocative response, expressing the complex emotional state of the person in the situation."### Analysis:- **D1** focuses on the young woman displaying expressions of surprise, shock, or anger, which do not align with the ground truth where she is described as being concerned and trying to console the crying girl.- **D2** contains a completely different context and description ("Our unfinished business") that does not match the scene described in the ground truth at all. It talks about the woman's emotional state being happy, excited, or curious, which contradicts the ground truth where she is concerned and trying to console the girl.### Conclusion: Neither of the provided predicted descriptions aligns well with the ground truth description. Since both descriptions contain significant discrepancies with the actual scene, the most appropriate response is:**same**This indicates that both descriptions are equally off the mark and do not accurately reflect the given ground truth.

---------------------------------------------------------------------------------------------------------------------------

Based on the provided description, please determine which one is better aligned with the ground truth description. The output should be 'D1', 'D2', or 'tie'.

Description: {Output of the Second Step}

**Output:** tie

Figure 23: Visualization of the internal outputs of S4.

## L    EMOTION RANGE ANALYSIS OF EMOPREFER-DATA

This paper focuses on *descriptive emotions*, which differ from *basic emotion recognition*. In *basic emotion recognition*, it restricts the label space to a few basic emotions. In contrast, *descriptive emotions* encompass an open-vocabulary set of emotion words. For clarification, Figure 24 visualizes descriptive emotions, which include diverse emotional words in an open-vocabulary manner. Figure 25 presents the emotion word cloud for EmoPrefer-Data, demonstrating that our dataset includes a wide range of emotions. Notably, we observe a skewed emotion distribution. For instance, *anger* appears more frequently than *helplessness*.

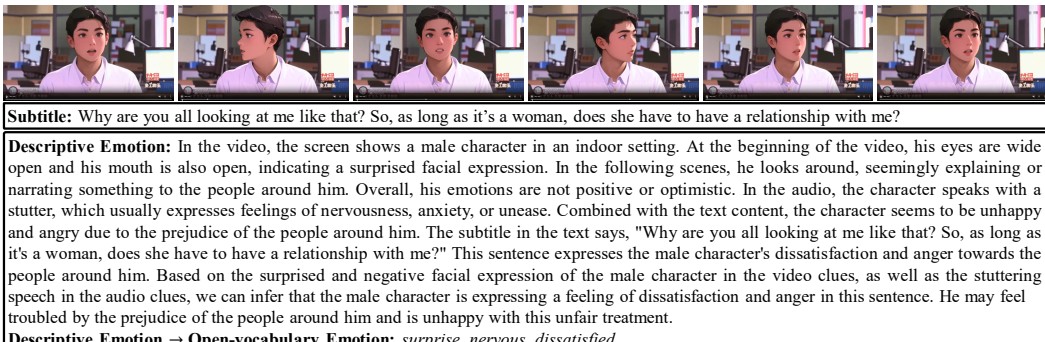

**Subtitle:** Why are you all looking at me like that? So, as long as it's a woman, does she have to have a relationship with me?

**Descriptive Emotion:** In the video, the screen shows a male character in an indoor setting. At the beginning of the video, his eyes are wide open and his mouth is also open, indicating a surprised facial expression. In the following scenes, he looks around, seemingly explaining or narrating something to the people around him. Overall, his emotions are not positive or optimistic. In the audio, the character speaks with a stutter, which usually expresses feelings of nervousness, anxiety, or unease. Combined with the text content, the character seems to be unhappy and angry due to the prejudice of the people around him. The subtitle in the text says, "Why are you all looking at me like that? So, as long as it's a woman, does she have to have a relationship with me?" This sentence expresses the male character's dissatisfaction and anger towards the people around him. Based on the surprised and negative facial expression of the male character in the video clues, as well as the stuttering speech in the audio clues, we can infer that the male character is expressing a feeling of dissatisfaction and anger in this sentence. He may feel troubled by the prejudice of the people around him and is unhappy with this unfair treatment.

**Descriptive Emotion → Open-vocabulary Emotion:** *surprise, nervous, dissatisfied*

Figure 24: Visualization of descriptive emotions, which include diverse emotional words in an open-vocabulary manner.

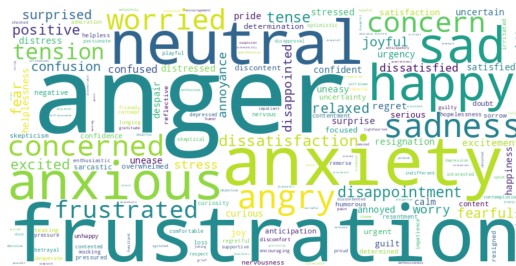

Figure 25: Emotion word cloud visualization of EmoPrefer-Data.

## M    IMPACT OF EMOTION DISTRIBUTION ON PREFERENCE

In this section, we investigate how emotion distribution influences human and MLLM preferences. Specifically, we conduct an insightful experiment. Figure 26 visualizes the emotion word clouds of preferred and non-preferred descriptions from both humans and MLLMs. These figures reveal similar distributions. For example, *anger* and *frustration* frequently appear in all distributions. To quantitatively assess their similarity, we compute the Jensen-Shannon Distance (JSD) between different distributions, where lower values indicate higher similarity. Since we do not restrict the emotion range in EmoPrefer-Data, we calculate this score only for overlapping emotions in two distributions. Experimental results in Figure 27 demonstrate that these distributions exhibit relatively high similarity. Based on both quantitative and qualitative analyses, we conclude that preferred and non-preferred descriptions from humans and MLLMs share similar emotion distributions. In other words, emotion distribution alone cannot predict human or MLLM preferences. For instance, the presence of *anger* does not reliably indicate preference. Therefore, we can conclude that emotion distribution does not introduce bias in MLLM's preference evaluation.

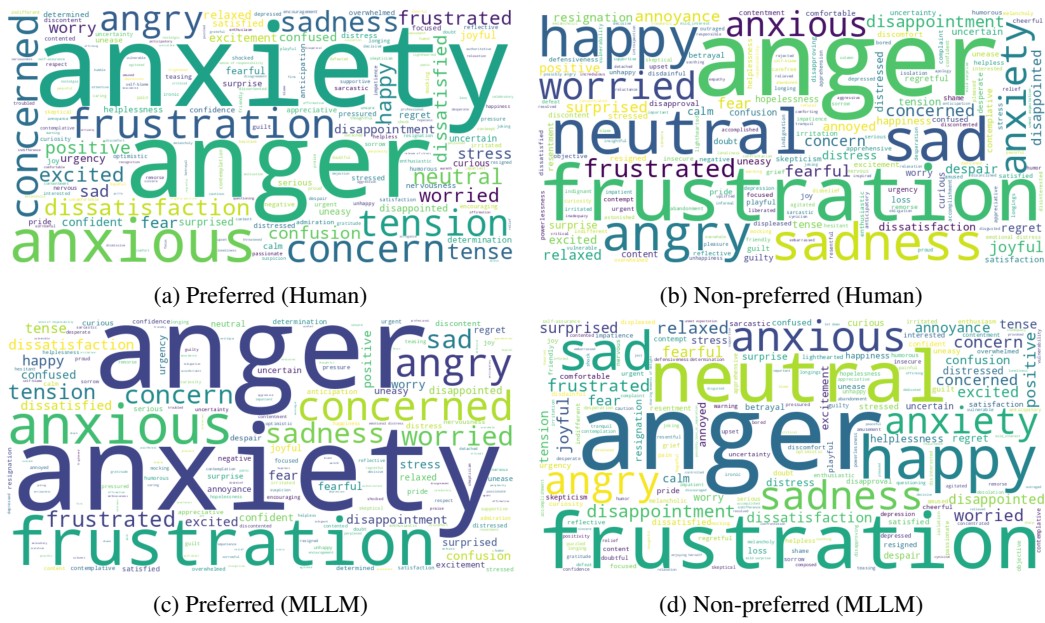

(a) Preferred (Human)  (b) Non-preferred (Human)

(c) Preferred (MLLM)  (d) Non-preferred (MLLM)

Figure 26: Word clouds for preferred and non-preferred descriptions of humans and MLLMs.

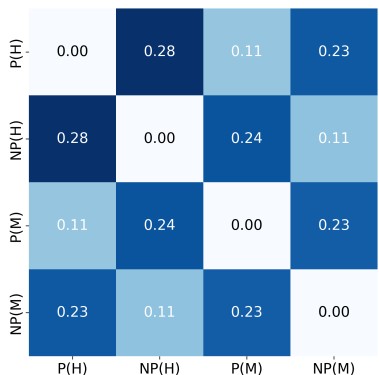

Figure 27: JSD between different emotion distributions, where lower values indicate higher similarity. *P(H)*, *NP(H)*, *P(M)*, and *NP(M)* are abbreviations for human-preferred, human-nonpreferred, MLLM-preferred, and MLLM-nonpreferred emotion distributions, respectively.

## N IMPACT OF TEMPLATES ON SWAP CONSISTENCY

In Figure 28, we examine three different templates and analyze their influence on *swap consistency*. Experimental results are presented in Table 6, where we use Qwen2.5-Omni as the MLLM. We observe that different templates lead to variations in *swap consistency*. Among them, template #2 achieves the best performance.

Table 6: Impact of prompt wording on *swap consistency*.

| Prompt Template | Swap Consistency |
|---|---|
| Template #1 | 79.97±0.00 |
| Template #2 | 87.80±0.00 |
| Template #3 | 80.31±0.00 |

---

*Template 1:*

We provide two descriptions for a given input. Please determine which one is better aligned with the input video. If both descriptions are equally aligned with the input content, please output 'tie'. Directly output the answer without additional reasoning.

Video: {Video}
D1: {Description_1}
D2: {Description_2}

-----------------------------------------------------------------------------------------------------------------

We provide two descriptions for a given input. Please determine which one is better aligned with the input video. If both descriptions are equally aligned with the input content, please output 'tie'. Directly output the answer without additional reasoning.

Video: {Video}
D1: {Description_2}
D2: {Description_1}

---

*Template 2:*

Two descriptions are provided below. Please identify which one aligns better with the input video. If they are equally aligned, respond with 'tie'. Output only answers and no explanation needed.

Video: {Video}
D1: {Description_1}
D2: {Description_2}

-----------------------------------------------------------------------------------------------------------------

Two descriptions are provided below. Please identify which one aligns better with the input video. If they are equally aligned, respond with 'tie'. Output only answers and no explanation needed.

Video: {Video}
D1: {Description_2}
D2: {Description_1}

---

*Template 3:*

For the provided input video, we present two candidate descriptions. Determine which description is more aligned with the input content. If neither is preferable, return 'tie'. Your response should be either 'D1', 'D2', or 'tie', with no further commentary.

Video: {Video}
D1: {Description_1}
D2: {Description_2}

-----------------------------------------------------------------------------------------------------------------

For the provided input video, we present two candidate descriptions. Determine which description is more aligned with the input content. If neither is preferable, return 'tie'. Your response should be either 'D1', 'D2', or 'tie', with no further commentary.

Video: {Video}
D1: {Description_2}
D2: {Description_1}

---

Figure 28: Visualization of different prompt templates for *swap consistency*.

## O    ABLATION STUDIES ON RECOGNITION PERFORMANCE

### O.1    IMPACT OF INPUT MODALITY

In Table 7, we analyze how different input modalities affect recognition performance. Experimental results demonstrate that incorporating lexical modality consistently improves performance. For instance, audio-text inputs outperform audio-only inputs. However, using all modalities (audio-video-text) does not yield better results than using either audio-text or video-text inputs.

Table 7: Impact of input modality on *recognition performance*.

| Input Modality | | | Rec. (2-Class) | | Rec. (3-Class) | |
|---|---|---|---|---|---|---|
| **Audio** | **Video** | **Text** | **WAF($\uparrow$)** | **ACC($\uparrow$)** | **WAF($\uparrow$)** | **ACC($\uparrow$)** |
| ✔ | ✗ | ✗ | 65.16±0.00 | 65.19±0.00 | 63.30±0.00 | 63.94±0.00 |
| ✔ | ✗ | ✔ | 69.09±0.36 | 69.09±0.36 | 66.29±0.53 | 65.85±0.52 |
| ✗ | ✔ | ✗ | 66.93±0.00 | 67.50±0.00 | 65.02±0.00 | 66.20±0.00 |
| ✗ | ✔ | ✔ | 68.92±0.61 | 69.18±0.62 | 66.27±0.46 | 65.24±0.44 |
| ✔ | ✔ | ✗ | 67.54±0.00 | 67.67±0.00 | 65.61±0.00 | 66.38±0.00 |
| ✔ | ✔ | ✔ | 68.34±0.00 | 68.38±0.00 | 66.38±0.00 | 67.07±0.00 |

## O.2 IMPACT OF PROMPT TEMPLATES

To evaluate the model's sensitivity to prompt templates, we conduct experiments using three different templates for S1∼S4. These templates are illustrated in Figures 29∼31.

---

**Template 1:**
We provide two descriptions for a given input. Please determine which one is better aligned with the input video. If both descriptions are equally aligned with the input content, please output 'tie'. Directly output the answer without additional reasoning.
Video: {Video}
D1: {Description_1}
D2: {Description_2}

**Template 2:**
Two descriptions are provided below. Please identify which one aligns better with the input video. If they are equally aligned, respond with 'tie'. Output only answers and no explanation needed.
Video: {Video}
D1: {Description_1}
D2: {Description_2}

**Template 3:**
For the provided input video, we present two candidate descriptions. Determine which description is more aligned with the input content. If neither is preferable, return 'tie'. Your response should be either 'D1', 'D2', or 'tie', with no further commentary.
Video: {Video}
D1: {Description_1}
D2: {Description_2}

---

Figure 29: Different prompt templates for S1.

*Template 1:*
Please provide a detailed description of a given video, with a particular focus on the emotions portrayed.
Video: {Video}

*Template 2:*
Please provide a detailed description of the given video, with a particular focus on the emotions expressed.
Video: {Video}

*Template 3:*
Describe the given video in detail, paying special attention to the emotions shown.
Video: {Video}

------------------------------------------------------------------------------------------------------------------------------

*Template 1:*
We provide a true description and two predicted descriptions. Please determine which one is better aligned with the true description. If both descriptions are equally aligned with the true description, please output 'tie'. Directly output the answer without additional reasoning.
Ground Truth: {Output of the First Step}
D1: {Description_1}
D2: {Description_2}

*Template 2:*
The ground truth description and two predicted descriptions are provided below. Please identify which predicted description aligns better with the ground truth. If both are equally aligned, output 'tie'. Your answer should be either 'D1', 'D2', or 'tie'. Do not include any reasoning.
Ground Truth: {Output of the First Step}
D1: {Description_1}
D2: {Description_2}

*Template 3:*
We provide the correct description and two model-generated descriptions. Determine which of the two predicted descriptions is more consistent with the ground truth. If there is no clear difference in alignment, output 'tie'. Please respond with 'D1', 'D2', or 'tie' only.
Ground Truth: {Output of the First Step}
D1: {Description_1}
D2: {Description_2}

Figure 30: Different prompt templates for S2 and S3.

*Template 1:*
Please provide a detailed description of a given video, with a particular focus on the emotions portrayed.
Video: {Video}

*Template 2:*
Please provide a detailed description of the given video, with a particular focus on the emotions expressed.
Video: {Video}

*Template 3:*
Describe the given video in detail, paying special attention to the emotions shown.
Video: {Video}

-----------------------------------------------------------------------------------------------------------------------

*Template 1:*
We provide a true description and two predicted descriptions. Please determine which one is better aligned with the true description. If both descriptions are equally aligned with the true description, please output 'tie'. Please output the answer along with the reasoning process.
Ground Truth: {Output of the First Step}
D1: {Description_1}
D2: {Description_2}

*Template 2:*
The ground truth description and two predicted descriptions are given. Please determine which of the two predicted descriptions is better aligned with the ground truth. If both are equally aligned, please output 'tie'. Provide your final answer ('D1', 'D2', or 'tie') along with the reasoning process that leads to your choice.
Ground Truth: {Output of the First Step}
D1: {Description_1}
D2: {Description_2}

*Template 3:*
Given the ground truth and two predicted description. Determine which predicted description is better aligned with the ground truth. If both are equally aligned, return 'tie'. Your output must include the final answer ('D1', 'D2', or 'tie') as well as a clear explanation of your reasoning.
Ground Truth: {Output of the First Step}
D1: {Description_1}
D2: {Description_2}

-----------------------------------------------------------------------------------------------------------------------

*Template 1:*
Based on the provided description, please determine which one is better aligned with the ground truth description. The output should be 'D1', 'D2', or 'tie'.
Description: {Output of the Second Step}

*Template 2:*
Based on the provided description, please determine which of the two options, 'D1' or 'D2', is better aligned with the ground truth description. The output should be either 'D1', 'D2', or 'tie'. Please directly output the answer without any reasoning.
Description: {Output of the Second Step}

*Template 3:*
Given the description, decide which is more aligned with the ground truth: 'D1' or 'D2'. If they are equally aligned, output 'tie'. Just output 'D1' or 'D2', or 'tie'. No reasoning needed.
Description: {Output of the Second Step}

Figure 31: Differnet prompt templates for S4.

As shown in Table 8, the impact of different prompt templates is minimal. Different prompt templates yield similar performance in emotion preference decoding.

Table 8: Impact of prompt templates on *recognition performance*.

| Template | Rec. (2-Class) | | Rec. (3-Class) | |
|---|---|---|---|---|
| | WAF(↑) | ACC(↑) | WAF(↑) | ACC(↑) |
| #1 | 67.74±0.00 | 67.85±0.00 | 65.81±0.00 | 66.55±0.00 |
| #2 | 68.02±0.00 | 68.03±0.00 | 66.07±0.00 | 66.72±0.00 |
| #3 | 66.46±0.00 | 66.61±0.00 | 64.57±0.00 | 65.33±0.00 |

## O.3 IMPACT OF TEMPERATURE AND SEED

Tables 9 and 10 present the recognition performance under varying temperatures and random seeds. For temperature, we test values in the range of 0.7 to 1.3, avoiding extreme settings. For random seeds, we compare three choices: 100, 200, and 300. Experimental results demonstrate that neither temperature nor seed affects recognition performance.

Table 9: Impact of temperature on *recognition performance*.

| Temperature | Rec. (2-Class) | | Rec. (3-Class) | |
|---|---|---|---|---|
| | WAF(↑) | ACC(↑) | WAF(↑) | ACC(↑) |
| 0.7 | 67.74±0.00 | 67.85±0.00 | 65.81±0.00 | 66.55±0.00 |
| 1.0 | 67.74±0.00 | 67.85±0.00 | 65.81±0.00 | 66.55±0.00 |
| 1.3 | 67.74±0.00 | 67.85±0.00 | 65.81±0.00 | 66.55±0.00 |

Table 10: Impact of seed on *recognition performance*.

| Seed | Rec. (2-Class) | | Rec. (3-Class) | |
|---|---|---|---|---|
| | WAF(↑) | ACC(↑) | WAF(↑) | ACC(↑) |
| 100 | 67.74±0.00 | 67.85±0.00 | 65.81±0.00 | 66.55±0.00 |
| 200 | 67.74±0.00 | 67.85±0.00 | 65.81±0.00 | 66.55±0.00 |
| 300 | 67.74±0.00 | 67.85±0.00 | 65.81±0.00 | 66.55±0.00 |

## P    EMOPREFER-DATA-V2

In **EmoPrefer-Data**, we recruited three annotators to label 1,368 pairs. In this section, we expanded the annotator pool to 14 annotators and labeled an additional 2,096 samples, creating a new dataset **EmoPrefer-Data-V2**. Table 11 presents the performance of various MLLMs on both datasets. We observe that Qwen2.5-Omni, which performed well on EmoPrefer-Data, also achieves strong performance on EmoPrefer-Data-V2. To quantitatively assess their similarity, we calculate the PCC similarity between corresponding metrics of the two datasets. In Table 11, the PCC score reaches above 0.75, a relatively high value. These results indicate that the conclusions drawn from EmoPrefer-Data are reflective of the MLLMs' statistical performance in emotion preference decoding.

Table 11: **Main results on EmoPrefer-Data and EmoPrefer-Data-V2.** This table reports the two-class recognition performance of various MLLMs under their best-performing prompting strategies.

| Model | EmoPrefer-Data | | EmoPrefer-Data-V2 | |
|---|---|---|---|---|
| | WAF($\uparrow$) | ACC($\uparrow$) | WAF($\uparrow$) | ACC($\uparrow$) |
| *Open-source MLLMs* | | | | |
| VideoChat (Li et al., 2025b) | 51.79$\pm$1.21 | 52.31$\pm$0.98 | 56.63$\pm$0.79 | 58.37$\pm$0.65 |
| Video-ChatGPT (Maaz et al., 2024) | 52.10$\pm$0.94 | 52.13$\pm$0.98 | 55.03$\pm$0.12 | 54.74$\pm$0.09 |
| Otter (Li et al., 2025a) | 52.12$\pm$1.10 | 52.49$\pm$0.98 | 58.53$\pm$0.12 | 61.14$\pm$0.15 |
| VideoChat2 (Li et al., 2024b) | 52.38$\pm$0.10 | 52.49$\pm$0.09 | 56.77$\pm$0.77 | 57.14$\pm$0.95 |
| mPLUG-Owl (Ye et al., 2023) | 53.76$\pm$0.17 | 53.82$\pm$0.18 | 56.66$\pm$1.40 | 56.15$\pm$1.45 |
| Video-LLaVA (Lin et al., 2024) | 54.53$\pm$0.81 | 54.62$\pm$0.80 | 56.50$\pm$0.25 | 56.98$\pm$0.25 |
| Chat-UniVi (Jin et al., 2024) | 54.97$\pm$1.64 | 55.06$\pm$1.60 | 55.38$\pm$0.15 | 55.35$\pm$0.28 |
| LLaVA-Next-Video (Li et al., 2024a) | 56.41$\pm$0.98 | 56.57$\pm$0.98 | 57.28$\pm$0.49 | 56.86$\pm$0.43 |
| LLaMA-VID (Li et al., 2024c) | 57.10$\pm$0.63 | 57.10$\pm$0.62 | 57.91$\pm$0.95 | 58.43$\pm$0.95 |
| PLLAVA (Xu et al., 2024) | 57.29$\pm$0.18 | 57.55$\pm$0.18 | 58.32$\pm$1.33 | 57.85$\pm$1.35 |
| VITA-1.5 (Fu et al., 2025) | 60.08$\pm$0.61 | 60.12$\pm$0.62 | 63.59$\pm$0.00 | 64.00$\pm$0.00 |
| Qwen2-Audio (Chu et al., 2024) | 63.17$\pm$0.19 | 63.23$\pm$0.18 | 58.53$\pm$0.27 | 58.98$\pm$0.28 |
| Qwen2.5-VL (Bai et al., 2025) | 64.43$\pm$0.87 | 65.28$\pm$0.80 | 61.66$\pm$0.61 | 62.12$\pm$0.65 |
| Qwen2.5-Omni (Xu et al., 2025) | 67.21$\pm$0.00 | 67.32$\pm$0.00 | 64.96$\pm$0.00 | 66.03$\pm$0.00 |
| *Closed-source MLLMs* | | | | |
| GPT-4o (OpenAI, 2024) | 59.28$\pm$0.08 | 59.41$\pm$0.09 | 64.54$\pm$0.00 | 64.62$\pm$0.00 |
| Gemini-2.0-Flash (Google, 2025a) | 59.80$\pm$0.88 | 60.39$\pm$0.89 | 63.53$\pm$0.50 | 63.69$\pm$0.55 |
| Gemini-2.5-Flash (Google, 2025b) | 60.60$\pm$0.64 | 61.19$\pm$0.62 | 65.06$\pm$0.37 | 65.60$\pm$0.43 |
| GPT-4.1 (Achiam et al., 2023) | 60.75$\pm$0.18 | 60.75$\pm$0.18 | 64.73$\pm$0.00 | 64.55$\pm$0.00 |
| Gemini-1.5-Pro (Team et al., 2024) | 60.79$\pm$0.68 | 61.55$\pm$0.62 | 61.94$\pm$0.06 | 61.42$\pm$0.06 |
| Gemini-1.5-Flash (Team et al., 2024) | 64.64$\pm$0.12 | 65.19$\pm$0.18 | 63.27$\pm$0.12 | 66.46$\pm$0.06 |
| *PCC Similarity Between Corresponding Metrics of Two Datasets* | | | | |
| PCC Score | 0.7754 | 0.7570 | 0.7754 | 0.7570 |

## Q  HUMAN-ONLY VS. HUMAN–LLM COLLABORATION

In Figure 32, we present an example illustrating the differences between human-only annotations and those generated through human–LLM collaboration. We observe that human–LLM collaboration produces more detailed and refined descriptions with the assistance of MLLMs.

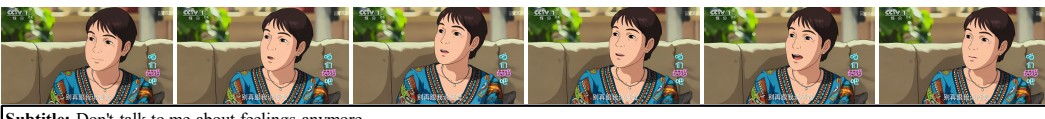

**Subtitle:** Don't talk to me about feelings anymore.

**Human-only Annotation:** In the video, she roared, spoke passionately, spoke quickly, spoke loudly, and stared with wide eyes. Therefore, it is inferred that her emotional state is anger.

**Human-LLM Collaboration:** In the video, we see a woman with expressive mouth movements, seemingly expressing her views or emotions more strongly. In the audio, the voice expresses anger. The tone is sharp and aggressive. There are no pauses, the speech is fast, and the volume is high. In the text, the subtitle says, ""Don't talk to me about feelings anymore."" This sentence may be the woman's response to someone or something. Based on the video clue of the woman's expressive mouth movements and the audio clue of an angry voice with a sharp and aggressive tone, fast speech, and high volume, we can infer that this sentence carries a sense of anger and dissatisfaction. The woman may be very unhappy with the other person's words or actions, and therefore expresses her emotions with an aggressive tone.

Figure 32: Comparing annotation quality: *human-only* vs. *human–LLM collaboration*.

