# OpenReview forum: "EmoPrefer: Can Large Language Models Understand Human Emotion Preferences?"
_ICLR.cc/2026/Conference — ICLR 2026 Poster_

### Official Review · Reviewer_yfmT · 2025-10-27

**Soundness:** 2
**Presentation:** 2
**Contribution:** 1
**Rating:** 2
**Confidence:** 2

**Summary:**

This paper introduces EmoPrefer, a benchmark designed to evaluate whether large language models can understand human emotion preferences.
The authors first construct a multimodal dataset, EmoPrefer-Data, by pairing two different emotion descriptions for the same video and collecting human preference annotations indicating which description better reflects the perceived emotion.
Based on this dataset, they develop EmoPrefer-Bench, which assesses the ability of multimodal large language models to act as “judges” of emotional preference—that is, to decide which of two emotion descriptions aligns more closely with human judgment.
The benchmark measures both recognition accuracy and swap consistency, testing whether a model’s decision is invariant to input order.

**Strengths:**

1. Novelty and scope
 Proposes a first benchmark targeting emotion preference with MLLMs-as-judges, plus a new multimodal preference dataset and a clear pipeline (data, bench, metrics).

2. Dataset curation
Uses overlapping videos from two descriptive MER sources, applies annotator screening and consensus filtering, and reports inter-annotator agreement; tie handling is explicitly discussed.

3. Evaluation design
Defines two complementary metrics—recognition performance (2-class/3-class WAF/ACC) and swap consistency—to capture both accuracy and order-invariance.

**Weaknesses:**

1. Potential LLM–LLM evaluation loop
The compared descriptions originate from model-generated datasets; humans provide pairwise preferences rather than authoring gold descriptions, risking measurement of model self-consistency more than human preference.

2. Limited human supervision
Small, filtered annotator pool and consensus-only retention reduce diversity; the reported human upper bound is modest and ties lower agreement further, constraining ceiling estimates.

3. Cost and scalability
No systematic reporting of inference cost across S1–S4, nor analysis of error accumulation vs. chain length or budgeted evaluation settings.

4. Missing ablations
No systematic per-modality or prompt-sensitivity ablations (e.g., audio-off, video-off, text-only), and limited analysis of temperature/seed effects for judge stability.

Overall, the paper addresses a novel and relevant topic that bridges affective computing and large model evaluation.
It provides a useful dataset and benchmark to analyze how well MLLMs capture subjective human emotional judgments.
However, since the emotion descriptions used for annotation originate from other model-generated datasets, the study risks forming a closed LLM–LLM evaluation loop, where models are effectively judging content generated by other models rather than real human descriptions. I think the author can reconsider the data construction process. It is difficult for the community to be convinced by the purely synthetic data benchmark.

**Questions:**

1. Inconsistent variable naming in the swap-consistency formula
In the definition of the swap-consistency metric (Equation (1)), the variables “onnormal” and “onswapped” are used inside the mathematical expression. These concatenated English words are non-standard and may confuse readers.

2. Unmatched quotation marks in the prompt template
In Figure 17 (Appendix D.3, “Details of S4”), the word 'Tie’ is enclosed with mismatched quotation marks—one straight quote and one curly quote.

3. Redundant wording in contribution statements
The sentence “Additionally, we introduce additional strategies ...” repeats the word “additional”.

---

> ### Author Response · Authors · 2025-11-21
> **Author Response (Part1)**
>
> Thank you for recognizing the strengths of this paper in terms of task importance, novelty, dataset, and evaluation metrics.
>
> *Q1: The compared descriptions originate from model-generated datasets; humans provide pairwise preferences rather than authoring gold descriptions, risking measurement of model self-consistency more than human preference. Since the emotion descriptions used for annotation originate from other model-generated datasets, the study risks forming a closed LLM–LLM evaluation loop, where models are effectively judging content generated by other models rather than real human descriptions. I think the author can reconsider the data construction process. It is difficult for the community to be convinced by the purely synthetic data benchmark.*
>
> A1: We believe the reviewer may have some misunderstandings about our work. Below, we address your questions from the following aspects:
>
> **(1) Description Source.** Our descriptions are sourced from two benchmark datasets: MER-Caption+ and MERR-Fine. We would like to clarify that these datasets are not entirely LLM-generated. To ensure high quality, human annotators are actively involved in their creation. For MER-Caption+, human priors are used to guide both description generation and sample filtering. For MERR-Fine, human annotators manually remove descriptions that do not align with the character’s emotional state. Thus, the annotation processes of both datasets incorporate human guidance, which helps ensure the quality and reliability of the emotion descriptions. This is also the key reason we selected descriptions from these datasets for preference annotation.
>
> **(2) Rationality.** As explained above, the descriptions are not fully automatically generated but involve human annotations. Even if they are fully synthetic, using preference annotations from synthetic datasets is a common and accepted practice in current leading preference benchmarks [1, 2]. Meanwhile, alignment training on such datasets has been shown to bring significant performance improvements across various domains [1].
>
> **(3) Annotation Content.** In EmoPrefer-Data, the ground-truth preferences are provided by humans, not generated by LLMs.
>
> [1] Zhang, YiFan, Tao Yu, Haochen Tian, Chaoyou Fu, Peiyan Li, Jianshu Zeng, Wulin Xie et al. "MM-RLHF: The Next Step Forward in Multimodal LLM Alignment." In the Forty-second International Conference on Machine Learning.
>
> [2] Chen, Dongping, Ruoxi Chen, Shilin Zhang, Yaochen Wang, Yinuo Liu, Huichi Zhou, Qihui Zhang, Yao Wan, Pan Zhou, and Lichao Sun. "Mllm-as-a-judge: Assessing multimodal llm-as-a-judge with vision-language benchmark." In the Forty-first International Conference on Machine Learning. 2024.
>
> *Q2: Small, filtered annotator pool and consensus-only retention reduce diversity; the reported human upper bound is modest and ties lower agreement further, constraining ceiling estimates.*
>
> A2: We would like to address your question from the following perspectives:
>
> **(1) Necessity of annotator filtering.** Emotions are a relatively complex internal state of human beings. To ensure the reliability of the annotation results, the process of annotator filtering is essential [1, 2]. This step helps to remove annotators who demonstrate insufficient attention to the annotation task or whose perception of emotions significantly deviates from that of the general population.
>
> **(2) Expand the pool of annotators and no longer rely solely on consensus.** In *EmoPrefer-Data*, we recruited three annotators to label 1,368 pairs. In response to your question, we expanded the annotator pool to 14 and additionally labeled 2,096 samples, retaining not only the consensus samples but also other non-consensus ones. This new dataset is denoted as *EmoPrefer-Data-V2*. More details can be found in Appendix P.
>
> [1] Li, Shan, Weihong Deng, and JunPing Du. "Reliable crowdsourcing and deep locality-preserving learning for expression recognition in the wild." In Proceedings of the IEEE conference on computer vision and pattern recognition, pp. 2852-2861. 2017.
>
> [2] Lian, Zheng, Haiyang Sun, Licai Sun, Kang Chen, Mingyu Xu, Kexin Wang, Ke Xu et al. "Mer 2023: Multi-label learning, modality robustness, and semi-supervised learning." In Proceedings of the 31st ACM International Conference on Multimedia, pp. 9610-9614. 2023.

---

> ### Author Response · Authors · 2025-11-21
> **Author Response (Part2)**
>
> *Q3: No systematic reporting of inference cost across S1–S4, nor analysis of error accumulation vs. chain length or budgeted evaluation settings.*
>
> A3: We would like to address your question from the following perspectives:
>
> **(1) Inference Cost Comparison.** We fix the MLLM to Qwen2.5-Omni and report the inference costs across S1$\sim$S4. As shown in Table 1, S1 is the fastest strategy due to its fewer inference steps. Meanwhile, S3/S4 runs faster than S2. The primary reason is that we leverage vLLM [1] to accelerate the inference process of external LLMs in S3/S4, resulting in reduced inference time.
>
> #### *Table 1. Inference cost analysis for different prompting strategies*
>
> | Prompting Strategy | Inference Time per Sample (seconds) |
> |--------------------|-------------------------------------|
> | S1                 | 1.3875                              |
> | S2                 | 5.4188                              |
> | S3                 | 4.0442                              |
> | S4                 | 4.9370                              |
>
>
> **(2) Error Accumulation Analysis.** We provide additional analysis on the correlation between error accumulation and chain length. The experimental results in Figure 6 show that for Qwen2.5-Omni, S1/S2 outperform S3/S4. The main reason is that longer chains in S3/S4 may increase the risk of error accumulation. However, for Chat-UniVi, we observe a different trend: S3/S4 perform better than S1/S2. This is because Chat-UniVi has limited language processing capabilities, and using external LLMs in S3/S4 helps the model better interpret user instructions and generate preference predictions. Thus, a trade-off emerges: while longer inference chains may raise the risk of error accumulation, the external LLMs in long chains (i.e., S3/S4) can compensate for the limited language processing abilities of MLLMs, ultimately leading to better performance. More analysis is provided in Appendix K.
>
> [1] Kwon, Woosuk, Zhuohan Li, Siyuan Zhuang, Ying Sheng, Lianmin Zheng, Cody Hao Yu, Joseph Gonzalez, Hao Zhang, and Ion Stoica. "Efficient memory management for large language model serving with pagedattention." In Proceedings of the 29th Symposium on Operating Systems Principles, pp. 611-626. 2023.

---

> > ### Author Response · Authors · 2025-11-21
> > **Author Response (Part3)**
> >
> > *Q4: No systematic per-modality or prompt-sensitivity ablations (e.g., audio-off, video-off, text-only), and limited analysis of temperature/seed effects for judge stability.*
> >
> > A4: We conduct ablation studies on input modalities, prompt sensitivity, the impact of temperature, and the effect of seeds. We use Qwen2.5-Omni for these ablation studies.
> >
> > **(1) Impact of Input Modality.** In Table 2, we analyze how different input modalities affect recognition performance. Experimental results demonstrate that incorporating lexical modality consistently improves performance. For instance, audio-text inputs outperform audio-only inputs. However, using all modalities (audio-video-text) does not yield better results than using either audio-text or video-text inputs.
> >
> > #### *Table 2. Impact of input modality on recognition performance.*
> >
> > | Audio | Video | Text | Rec. (2-Class)       |              | Rec. (3-Class)     |                |
> > |-------|-------|------|-----|----|-----|----|
> > |       |       |      | **WAF(↑)** | **ACC(↑)** | **WAF(↑)** | **ACC(↑)** |
> > | ✓     | ✗     | ✗    | 65.16±0.00 | 65.19±0.00 | 63.30±0.00 | 63.94±0.00 |
> > | ✓     | ✗     | ✓    | 69.09±0.36 | 69.09±0.36 | 66.29±0.53 | 65.85±0.52 |
> > | ✗     | ✓     | ✗    | 66.93±0.00 | 67.50±0.00 | 65.02±0.00 | 66.20±0.00 |
> > | ✗     | ✓     | ✓    | 68.92±0.61 | 69.18±0.62 | 66.27±0.46 | 65.24±0.44 |
> > | ✓     | ✓     | ✗    | 67.54±0.00 | 67.67±0.00 | 65.61±0.00 | 66.38±0.00 |
> > | ✓     | ✓     | ✓    | 68.34±0.00 | 68.38±0.00 | 66.38±0.00 | 67.07±0.00 |
> >
> > **(2) Impact of Prompt Templates.** To evaluate the model's sensitivity to prompt templates, we conduct experiments using three different templates for S1$\sim$S4. These templates are illustrated in Figures 29$\sim$31 (in the updated PDF). As shown in Table 3, the impact of different prompt templates is minimal. Different prompt templates yield similar performance in emotion preference decoding.
> >
> > #### *Table 3. Impact of prompt templates on recognition performance.*
> >
> > | Template | Rec. (2-Class)        |             | Rec. (3-Class)            |         |
> > |----------|------------------------------------|------------------------------------|------------------------------------|------------------------------------|
> > |          | **WAF(↑)** | **ACC(↑)** | **WAF(↑)** | **ACC(↑)** |
> > | \#1      | 67.74±0.00 | 67.85±0.00 | 65.81±0.00 | 66.55±0.00 |
> > | \#2      | 68.02±0.00 | 68.03±0.00 | 66.07±0.00 | 66.72±0.00 |
> > | \#3      | 66.46±0.00 | 66.61±0.00 | 64.57±0.00 | 65.33±0.00 |
> >
> >
> > **(3) Impact of Temperature and Seed.** Tables 4 and 5 present the recognition performance under varying temperatures and random seeds. For temperature, we test values in the range of 0.7 to 1.3, avoiding extreme settings. For random seeds, we compare three choices: 100, 200, and 300. Experimental results demonstrate that neither temperature nor seed affects recognition performance.
> >
> > #### *Table 4. Impact of temperature on recognition performance.*
> >
> > | Temperature | Rec. (2-Class)          |           | Rec. (3-Class)    |                 |
> > |-------------|------------------------------------|------------------------------------|------------------------------------|------------------------------------|
> > |             | **WAF(↑)** | **ACC(↑)** | **WAF(↑)** | **ACC(↑)** |
> > | 0.7         | 67.74±0.00 | 67.85±0.00 | 65.81±0.00 | 66.55±0.00 |
> > | 1.0         | 67.74±0.00 | 67.85±0.00 | 65.81±0.00 | 66.55±0.00 |
> > | 1.3         | 67.74±0.00 | 67.85±0.00 | 65.81±0.00 | 66.55±0.00 |
> >
> > #### *Table 5. Impact of seed on recognition performance*
> >
> > | Seed  | Rec. (2-Class)                |     | Rec. (3-Class)   |                  |
> > |-------|------------------------------------|------------------------------------|-|-|
> > |       | **WAF(↑)** | **ACC(↑)** | **WAF(↑)** | **ACC(↑)** |
> > | 100   | 67.74±0.00 | 67.85±0.00 | 65.81±0.00 | 66.55±0.00 |
> > | 200   | 67.74±0.00 | 67.85±0.00 | 65.81±0.00 | 66.55±0.00 |
> > | 300   | 67.74±0.00 | 67.85±0.00 | 65.81±0.00 | 66.55±0.00 |
> >
> > *Q5: In the definition of the swap-consistency metric (Equation (1)), the variables “onnormal” and “onswapped” are used inside the mathematical expression. These concatenated English words are non-standard and may confuse readers.*
> >
> > A5: In the revised manuscript, we have replaced $o^n_ {\text{normal}}$ and $o^n_ {\text{swapped}}$ with $o^n_ {\text{12}}$ and $o^n_ {\text{21}}$.
> >
> > *Q6: Unmatched quotation marks in the prompt template. In Figure 17 (Appendix D.3, “Details of S4”), the word 'Tie’ is enclosed with mismatched quotation marks—one straight quote and one curly quote.*
> >
> > A6: In the revised manuscript, we have corrected the unmatched quotation mark.
> >
> > *Q7: Redundant wording in contribution statements. The sentence “Additionally, we introduce additional strategies ...” repeats the word “additional”.*
> >
> > A7: In the revised manuscript, we have revised this statement.

---

> > > ### Comment · Reviewer_yfmT · 2025-11-24
> > > **Thank you for your reply.**
> > >
> > > My question has been initially clarified, thank you for your reply. I have increased my rating. Furthermore, I sincerely hope you can consider optimizing the data construction process and incorporating more human involvement, such as starting the construction from human descriptions and then filtering them using LLM. I still believe that constructing datasets using an LLM-based synthetic approach is a rather worrying practice.

---

> ### Author Response · Authors · 2025-11-25
> **Author Response**
>
> Thanks for raising your scores. We would like to address your question from the following four perspectives:
>
> **(1) Benefits of Human–LLM Collaboration.** MERR-Fine and MER-Caption+ are not synthetic datasets created without human involvement; instead, they leverage human–LLM collaboration to ensure high-quality annotation results. This collaborative approach yields richer and more nuanced descriptions, which are generally preferred by human evaluators, an observation supported by prior work [1]. In Figure 32 (in the updated PDF), we present an example illustrating the differences between human-only annotations and those generated through human–LLM collaboration. We observe that human–LLM collaboration produces more detailed and refined descriptions with the assistance of MLLMs. **Therefore, human–LLM collaboration ensures higher annotation quality than human-only annotation.**
>
> **(2) Development of Descriptive Emotion Datasets.** Emotions are inherently linked to a wide range of human behaviors, including facial expressions, (micro-)gestures, head movements, hand actions, and vocal tones. As a result, generating a comprehensive and accurate description of a person’s emotional state solely through human annotators is inherently challenging. In the early stages, MAFW [2] was the first dataset to provide descriptive emotion labels, relying exclusively on human annotators to generate emotion descriptions. *However, human annotators tend to focus on the most salient cues, which can lead to incomplete descriptions and an omission of subtle, yet relevant, emotional signals. Furthermore, scaling such datasets is challenging.* To overcome this limitation, researchers began incorporating MLLMs to assist in description generation. This led to the construction of MERR-Fine [3], a human-led model-assisted dataset, and MER-Caption+ [4], a model-led human-assisted dataset. Researchers have observed that using MLLMs to assist in description generation results in richer, more nuanced descriptions that are generally preferred by humans, ultimately yielding higher-quality annotations than those produced by humans alone [1]. **This is why the human-MLLM cooperative strategy has become the mainstream approach for constructing descriptive emotion datasets.**
>
> **(3) Mainstream Preference Datasets in Other Fields.** Both MERR-Fine and MER-Caption+ rely on human–MLLM collaboration for annotation, rather than synthetic datasets generated without human involvement. In other words, even if such datasets are synthetic, annotating preferences on synthetic data is a widely adopted and accepted practice in mainstream preference benchmarks [5-9]. Recent studies have also shown that alignment training on these datasets can significantly improve performance [5-9].
>
> **(4) Human-based Preference Annotation.** We would like to emphasize that this paper focuses on emotion preference rather than emotion description. Regardless of the source of the compared description pairs, all preference labels are annotated by humans. These human-annotated preference results reflect people’s preferences across different emotion representations.
>
> [1] "OV-MER: Towards Open-Vocabulary Multimodal Emotion Recognition." In the Forty-second International Conference on Machine Learning.
>
> [2] "Mafw: A large-scale, multi-modal, compound affective database for dynamic facial expression recognition in the wild." In Proceedings of the 30th ACM International Conference on Multimedia, pp. 24-32. 2022.
>
> [3] "Emotion-llama: Multimodal emotion recognition and reasoning with instruction tuning." Advances in Neural Information Processing Systems 37 (2024): 110805-110853.
>
> [4] "AffectGPT: A New Dataset, Model, and Benchmark for Emotion Understanding with Multimodal Large Language Models." In the Forty-second International Conference on Machine Learning.
>
> [5] "MM-RLHF: The Next Step Forward in Multimodal LLM Alignment." In the Forty-second International Conference on Machine Learning.
>
> [6] "Mllm-as-a-judge: Assessing multimodal llm-as-a-judge with vision-language benchmark." In the Forty-first International Conference on Machine Learning. 2024.
>
> [7] "Multimodal rewardbench: Holistic evaluation of reward models for vision language models." arXiv preprint arXiv:2502.14191 (2025).
>
> [8] "VLFeedback: A Large-Scale AI Feedback Dataset for Large Vision-Language Models Alignment." In Proceedings of the 2024 Conference on Empirical Methods in Natural Language Processing, pp. 6227-6246. 2024.
>
> [9] "VL-RewardBench: A Challenging Benchmark for Vision-Language Generative Reward Models." In Proceedings of the Computer Vision and Pattern Recognition Conference, pp. 24657-24668. 2025.

---

### Official Review · Reviewer_evbf · 2025-10-31

**Soundness:** 3
**Presentation:** 3
**Contribution:** 3
**Rating:** 6
**Confidence:** 4

**Summary:**

This paper addresses a gap in Descriptive Multimodal Emotion Recognition: the lack of cost-efficient, reliable evaluation methods for open-ended emotion descriptions. By proposing EmoPrefer—a framework that leverages Multimodal Large Language Models to decode human emotion preferences—the work makes three notable contributions: (1) EmoPrefer-Data, the first multimodal dataset for emotion preference annotation with expert consensus; (2) EmoPrefer-Bench, a benchmark to evaluate MLLMs/prompting strategies for preference prediction; and (3) empirical insights into MLLM behavior.

**Strengths:**

- High-Quality Dataset Construction. EmoPrefer-Data addresses common pitfalls in preference datasets.
- EmoPrefer-Bench is well-designed.
- Two complementary metrics—recognition performance (alignment with human annotations) and swap consistency (robustness to input order)—address both accuracy and reliability, a rare focus in LLM-as-judge work.

**Weaknesses:**

- The paper reports 1,368 video samples, but it is unclear how many final annotated pairs are retained after filtering for annotator consensus. For example, if ties (which have lower agreement, 59.23%) are excluded, the effective dataset size may be smaller—this impacts the statistical power of MLLM evaluations.
- The paper does not specify the range of emotions in EmoPrefer-Data (e.g., is it balanced across positive/negative/neutral, or dominated by common emotions like anxiety/frustration?). A skewed emotion distribution could bias MLLM evaluation.

**Questions:**

See weaknesses

---

> ### Author Response · Authors · 2025-11-21
> **Author Response (Part1)**
>
> Thank you for recognizing the importance of our dataset, benchmark, and evaluation metrics. We also appreciate your positive feedback on the soundness, presentation, and overall contribution of our work.
>
> *Q1: The paper reports 1,368 video samples, but it is unclear how many final annotated pairs are retained after filtering for annotator consensus. For example, if ties (which have lower agreement, 59.23\%) are excluded, the effective dataset size may be smaller—this impacts the statistical power of MLLM evaluations.*
>
> A1: In *EmoPrefer-Data*, we recruited three annotators to label 1,368 pairs. To ensure label reliability, we only retained samples with consensus preferences, resulting in a final dataset containing fewer than 1,368 samples. In response to your concern regarding whether the conclusions drawn from this dataset adequately reflect the statistical power of MLLMs, we expanded the annotator pool to 14 annotators and labeled an additional 2,096 samples, creating a new dataset *EmoPrefer-Data-V2*. Table 1 presents the performance of various MLLMs on both datasets. We observe that Qwen2.5-Omni, which performed well on EmoPrefer-Data, also achieves strong performance on EmoPrefer-Data-V2. To quantitatively assess their similarity, we calculate the PCC similarity between corresponding metrics of the two datasets. In Table 1, the PCC score reaches above 0.75, a relatively high value. *The performance similarity indicates that the conclusions drawn from EmoPrefer-Data are reflective of the MLLMs' statistical performance in emotion preference decoding.*
>
> #### *Table 1. Main results on EmoPrefer-Data and EmoPrefer-Data-V2. This table reports the two-class recognition performance of various MLLMs under their best-performing prompting strategies.*
>
>
> | Model  | EmoPrefer-Data|| EmoPrefer-Data-V2||
> |-|-----|-----|-----|-----|
> | | WAF(↑)| ACC(↑)| WAF(↑)| ACC(↑)|
> | **Open-source MLLMs** |||||
> | VideoChat | 51.79±1.21| 52.31±0.98| 56.63±0.79| 58.37±0.65|
> | Video-ChatGPT | 52.10±0.94| 52.13±0.98| 55.03±0.12| 54.74±0.09|
> | Otter  | 52.12±1.10| 52.49±0.98| 58.53±0.12| 61.14±0.15|
> | VideoChat2| 52.38±0.10| 52.49±0.09| 56.77±0.77| 57.14±0.95|
> | mPLUG-Owl | 53.76±0.17| 53.82±0.18| 56.66±1.40| 56.15±1.45|
> | Video-LLaVA   | 54.53±0.81| 54.62±0.80| 56.50±0.25| 56.98±0.25|
> | Chat-UniVi| 54.97±1.64| 55.06±1.60| 55.38±0.15| 55.35±0.28|
> | LLaVA-Next-Video | 56.41±0.98| 56.57±0.98| 57.28±0.49| 56.86±0.43|
> | LLaMA-VID | 57.10±0.63| 57.10±0.62| 57.91±0.95| 58.43±0.95|
> | PLLAVA | 57.29±0.18| 57.55±0.18| 58.32±1.33| 57.85±1.35|
> | VITA-1.5  | 60.08±0.61| 60.12±0.62| 63.59±0.00| 64.00±0.00|
> | Qwen2.5-VL| 64.43±0.87| 65.28±0.80| 61.66±0.61| 62.12±0.65|
> | Qwen2.5-Omni  | 67.21±0.00| 67.32±0.00| 64.96±0.00| 66.03±0.00|
> | |||||
> | **Closed-source MLLMs** |||||
> | GPT-4o | 59.28±0.08| 59.41±0.09| 64.54±0.00| 64.62±0.00|
> | Gemini-2.0-Flash | 59.80±0.88| 60.39±0.89| 63.53±0.50| 63.69±0.55|
> | Gemini-2.5-Flash | 60.60±0.64| 61.19±0.62| 65.06±0.37| 65.60±0.43|
> | GPT-4.1| 60.75±0.18| 60.75±0.18| 64.73±0.00| 64.55±0.00|
> | Gemini-1.5-Pro| 60.79±0.68| 61.55±0.62| 61.94±0.06| 61.42±0.06|
> | Gemini-1.5-Flash | 64.64±0.12| 65.19±0.18| 63.27±0.12| 66.46±0.06|
> | |||||
> | **PCC Similarity Between Corresponding Metrics of Two Datasets** |||||
> | PCC Score | 0.7754| 0.7570| 0.7754| 0.7570|

---

> ### Author Response · Authors · 2025-11-21
> **Author Response (Part2)**
>
> *Q2: The paper does not specify the range of emotions in EmoPrefer-Data (e.g., is it balanced across positive/negative/neutral, or dominated by common emotions like anxiety/frustration?). A skewed emotion distribution could bias MLLM evaluation.*
>
> A2: We would like to address your question from two perspectives:
>
> **(1) Emotion Range Analysis of EmoPrefer-Data.** This paper focuses on *descriptive emotions*, which differ from *basic emotion recognition*. In *basic emotion recognition*, it restricts the label space to a few basic emotions. In contrast, *descriptive emotions* encompass an open-vocabulary set of emotion words. For clarification, Figure 24 (in the updated PDF) visualizes descriptive emotions, which include diverse emotional words in an open-vocabulary manner. Figure 25 (in the updated PDF) presents the emotion word cloud for EmoPrefer-Data, demonstrating that our dataset includes a wide range of emotions.
>
> **(2) Impact of Emotion Distribution on Preference.** Then, we investigate how emotion distribution influences human and MLLM preferences. Specifically, we conduct an insightful experiment. Figure 26 (in the updated PDF) visualizes the emotion word clouds of preferred and non-preferred descriptions from both humans and MLLMs. These figures reveal similar distributions. For example, *anger* and *frustration* frequently appear in all distributions. To quantitatively assess their similarity, we compute the Jensen-Shannon Distance between different distributions, where lower values indicate higher similarity. Since we do not restrict the emotion range in EmoPrefer-Data, we calculate this score only for overlapping emotions in two distributions. Experimental results in Figure 27 (in the updated PDF) demonstrate that these distributions exhibit relatively high similarity. Based on both quantitative and qualitative analyses, we conclude that preferred and non-preferred descriptions from humans and MLLMs share similar emotion distributions. In other words, *emotion distribution alone cannot predict human or MLLM preferences*. For instance, the presence of *anger* does not reliably indicate preference. *Therefore, we can conclude that emotion distribution does not introduce bias in MLLM's preference evaluation.*

---

### Official Review · Reviewer_y5v9 · 2025-10-31

**Soundness:** 3
**Presentation:** 3
**Contribution:** 3
**Rating:** 6
**Confidence:** 3

**Summary:**

This paper proposed EmoPrefer, which presents a pioneering exploration into whether MLLMs can serve as cost-effective evaluators of human emotional understanding.
This work constructed EmoPrefer-Data, EmoPrefer-Bench, and evaluation metrics. These will contribute to the study of this area.
The experiments show that open-source models such as Qwen2.5-Omni can rival or surpass closed-source ones (GPT-4.1, Gemini 2.5), achieving 67.2% WAF and 79.1% swap consistency. Model-based crowdsourcing further improves both performance and robustness. The authors argue this lays groundwork for emotion-aware reward modeling and emotionally intelligent AI.

**Strengths:**

1. This paper is working on a novel research problem. It also presents a EmoPrefer-Data and EmoPrefer-Bench to support the research for this area.
2. Experimental Setup is comprehensive with insightful analyses.

**Weaknesses:**

1. Limited Dataset Scale and Diversity. Only 1,368 annotated pairs. The videos are only sourced from MER2024.
2. Lack of Downstream Evaluation. No demonstration of how emotion preference prediction could enhance practical applications.

**Questions:**

1. Is “swap consistency” sensitive to prompt wording or model randomness?
2. Could the authors elaborate on how cultural bias in EmoPrefer-Data might affect model generalization?

---

> ### Author Response · Authors · 2025-11-21
> **Author Response (Part1)**
>
> Thank you for recognizing the importance of EmoPrefer and acknowledging that our EmoPrefer-Data and EmoPrefer-Bench can support the development of this field. We also appreciate your positive feedback regarding the soundness of our experiments, the clarity of our presentation, and the contributions of our work.
>
> *Q1: Limited Dataset Scale and Diversity. Only 1,368 annotated pairs. The videos are only sourced from MER2024.*
>
> A1: We would like to address your questions from the following aspects:
>
> **(1) Video diversity.** The primary reason we selected MER2024 as the source dataset is its large scale, containing over 115K video samples across diverse scenarios. As illustrated in Figure 13, the dataset includes both single-person videos (the first two examples) and multi-speaker interaction videos (the last example). Moreover, these videos take place in a variety of real-world settings. For instance, the first video occurs in a hospital, while the last one takes place at home. Therefore, MER2024 inherently encompasses a wide range of scenarios.
>
> **(2) Dataset expansion.** The original EmoPrefer-Data provides annotations for 1,368 description pairs. In response to your concerns, we have additionally annotated 2,096 samples by hiring 14 annotators. We refer to this expanded dataset as *EmoPrefer-Data-V2*. More details can be found in Appendix P.
>
> We would like to emphasize that this is the first work to explore the potential of MLLMs in the domain of emotion preference. It introduces the first emotion preference dataset, proposes comprehensive evaluation metrics, and establishes a benchmark for this field. We sincerely hope the reviewers will consider accepting this paper, allowing it to reach a wider audience. This work presents pioneering research that reveals the capabilities of MLLMs in understanding human emotion preferences. Beyond evaluation, the dataset and insights derived from our benchmark will facilitate the development of emotion-intelligent MLLMs.
>
>
> *Q2: Lack of Downstream Evaluation. No demonstration of how emotion preference prediction could enhance practical applications.*
>
> A2: Emotion plays a pivotal role in both recommendation systems and human-computer interaction. In recommendation systems, we can leverage users' emotional cues to infer their interest in a product, thereby helping to provide more relevant recommendations. In human-computer interaction, understanding and responding to users' emotions enables more natural, empathetic, and personalized interactions. Emotion-aware systems can adapt content, tone, or functionality based on user mood, thereby improving engagement, accessibility, and satisfaction.
>
> This paper focuses on emotion preference, a critical factor in the development of artificial intelligence. Specifically, emotion preference can be used for post-training MLLMs. For example, we can utilize our dataset and insights from our benchmark to train reward models and apply reinforcement learning to maximize these rewards, thereby enhancing MLLMs’ understanding of human emotions.
>
> *Q3: Is “swap consistency” sensitive to prompt wording or model randomness?*
>
> A3: We investigate the impact of prompt wording and model randomness on swap consistency.
>
> **(1) Impact of prompt wording.** In Figure 28 (in the updated PDF), we examine three different templates and analyze their influence on *swap consistency*. Experimental results are presented in Table 1. We observe that different templates lead to variations in *swap consistency*. Among them, template \#2 achieves the best performance.
>
> #### *Table1. Impact of prompt wording on swap consistency*
>
> | Prompt Template | Swap Consistency |
> |-----------------|------------------|
> | Template \#1 | 79.97 ± 0.00 |
> | Template \#2 | 87.80 ± 0.00 |
> | Template \#3 | 80.31 ± 0.00 |
>
> **(2) Impact of model randomness.** Table 2 illustrates the effect of model randomness on swap consistency. Specifically, we run each experiment twice and report both the average scores and the standard deviation. We observe that the standard deviation is small, suggesting that model randomness has a minimal impact on swap consistency.
>
> #### *Table 2. Impact of model randomness on swap consistency*
>
> | Prompting Strategy | Swap Consistency |
> |--------------------|------------------|
> | S1 | 73.87 ± 0.00 |
> | S2 | 79.09 ± 0.00 |
> | S3 | 72.65 ± 0.52 |
> | S4 | 67.94 ± 0.52 |
>
> *Q4: Could the authors elaborate on how cultural bias in EmoPrefer-Data might affect model generalization?*
>
> A4: Good suggestion. Cultural bias is a significant research topic in the field of affective computing. Currently, we are collaborating with research teams from other countries on human emotion preferences, and this topic will be discussed in our future paper. We appreciate your interest and hope you will continue to follow our subsequent research.

---

### Official Review · Reviewer_uypZ · 2025-10-31

**Soundness:** 3
**Presentation:** 3
**Contribution:** 3
**Rating:** 4
**Confidence:** 4

**Summary:**

This paper addresses the challenges in Descriptive Multimodal Emotion Recognition (DMER), which uses free-form natural language to describe emotions for finer-grained representation. To tackle evaluation difficulties and reduce the manual effort in preference annotation, the authors propose EmoPrefer, the first exploration of leveraging multimodal large language models (MLLMs) for emotion preference decoding. They introduce EmoPrefer-Data, a high-quality expert-annotated preference dataset, and EmoPrefer-Bench, a benchmark to evaluate MLLMs and prompting techniques.

**Strengths:**

The results of the experiment are promising.

**Weaknesses:**

1. The dataset's sample source is too narrow and the scenarios are too limited. All original videos are from the MER2024 dataset, and the scenarios are restricted to "a single character facing forward with complete audio," lacking diverse scenarios such as multi-person interaction and complex environmental background interference.
2. This paper does not explain the theoretical basis for choosing binary WAF as the default metric. Further correlation analysis between binary and tri-class performance should be provided (e.g., whether the performance rankings of different MLLMs are consistent under both settings) to confirm that the default metric does not distort model comparisons.
3. Exchange consistency only quantifies whether the results are consistent before and after the order is swapped, but it does not explore the root cause of order bias. For example, is this bias caused by the model's attention mechanism or the modality fusion logic?
4. The paper designs four prompting strategies (S1-S4) with increasing reasoning chain complexity, but does not clarify why external LLMs improve performance in S3/S4. It only attributes it to "MLLMs may have limited language understanding," but does not verify whether the improvement comes from the external LLM’s stronger language reasoning ability or the multi-step reasoning’s error reduction.

**Questions:**

See Weaknesses.

---

> ### Author Response · Authors · 2025-11-21
> **Author Response (Part1)**
>
> Thank you for your acknowledgment of our work in terms of soundness, presentation, and contributions, as well as your recognition of the task’s novelty and the high quality of our dataset.
>
> *Q1: The dataset's sample source is too narrow and the scenarios are too limited. All original videos are from the MER2024 dataset, and the scenarios are restricted to "a single character facing forward with complete audio," lacking diverse scenarios such as multi-person interaction and complex environmental background interference.*
>
> A1: We would like to emphasize that MER2024 is a large-scale dataset, containing over 115K samples with multi-speaker interactions and diverse scenarios. Below, we address your concerns from the following aspects:
>
> **(1) Existence of multi-person interaction videos.** We would like to clarify that MER2024 does not restrict videos to single-person scenarios. For example, consider the last row in Figure 13: the video shows one character facing the camera, who is interacting with another character facing away from the camera. Thus, MER2024 also includes multi-person interaction videos.
>
> **(2) Diversity of scenarios.** MER2024 contains a wide variety of scenarios. For instance, the first row in Figure 13 takes place in a hospital, while the last row occurs at home. Although MER2024 does not annotate video scenarios, our preliminary review suggests that the dataset includes videos from diverse scenarios.
>
> **(3) Expansion of EmoPrefer-Data.** To enrich the dataset, we annotate an additional 2,096 samples by 14 annotators. We refer to this expanded dataset as EmoPrefer-Data-V2. More details can be found in Appendix P.
>
> **(4) Future work.** Currently, MER2024 includes up to two-speaker interactions. In the future, we plan to broaden the dataset by incorporating three or more speakers. Additionally, the current dataset primarily focuses on short videos (3s$\sim$10s) [1]. In the future, we will include longer videos (e.g., over 60s) for analysis.
>
> In conclusion, we have analyzed two-speaker interaction data from diverse settings. Additionally, we have expanded EmoPrefer-Data by annotating more videos. Future work will extend this study to include interactions among three or more speakers and longer videos. Importantly, these extensions do not affect the core contributions of this paper: (1) it is the first work to explore the potential of MLLMs in emotion preference, (2) it introduces the first preference dataset, proposes comprehensive evaluation metrics, and establishes the first benchmark for this field.
>
> [1] Lian, Zheng, Haiyang Sun, Licai Sun, Zhuofan Wen, Siyuan Zhang, Shun Chen, Hao Gu et al. "Mer 2024: Semi-supervised learning, noise robustness, and open-vocabulary multimodal emotion recognition." In Proceedings of the 2nd International Workshop on Multimodal and Responsible Affective Computing, pp. 41-48. 2024.
>
> *Q2: This paper does not explain the theoretical basis for choosing binary WAF as the default metric. Further correlation analysis between binary and tri-class performance should be provided (e.g., whether the performance rankings of different MLLMs are consistent under both settings) to confirm that the default metric does not distort model comparisons.*
>
> A2: We would like to address your concerns from both theoretical and experimental perspectives:
>
> **(1) Theoretical Basis.** First, we opt for binary classification rather than tri-class because ties contain higher ambiguity than clear preferences (e.g., when one description is distinctly better than another). Binary performance focuses on more definitively labeled comparisons, enabling more reliable conclusions. Second, we chose WAF over ACC due to the inherent class imbalance in EmoPrefer-Data. As shown in Table 1, the $d_ 1$ category slightly outnumbers the $d_ 2$ category. In imbalanced datasets, ACC is dominated by the majority class, whereas WAF better accounts for class imbalance. Therefore, we adopted WAF as our primary metric. **Combining these two considerations, we use binary WAF as the default evaluation metric.**
>
> #### *Table1. Class distribution in EmoPrefer-Data. For a given video and two descriptions $(x, d_1, d_2)$, the preference label falls into one of three categories: $d_ 1$, $d_ 2$, or tie.*
>
> | Preference | Percentage |
> |------------|------------|
> | $d_ 1$      | 50.17%     |
> | $d_ 2$      | 47.91%     |
> | tie        | 1.92%      |
>
> **(2) Experimental Analysis.** We also report the correlation between different metrics. As shown in Figure 20 (in the updated PDF), PCC scores between these metrics are relatively high, indicating strong inter-metric correlations. Meanwhile, we examine the ranking consistency across metrics. Although the rankings from different metrics are not identical, their overall trends are similar. For instance, Qwen2.5-Omni consistently performs best in both binary and tri-class metrics, whereas VideoChat ranks relatively poorly across all metrics.

---

> > ### Author Response · Authors · 2025-11-21
> > **Author Response (Part2)**
> >
> > *Q3: Exchange consistency only quantifies whether the results are consistent before and after the order is swapped, but it does not explore the root cause of order bias. For example, is this bias caused by the model's attention mechanism or the modality fusion logic?*
> >
> > A3: Good question, an interesting point I hadn’t previously considered in depth. To address your concern, we specifically designed an experiment to investigate the causes of order bias. We divided all samples into two subsets: (1) *those with order bias* and (2) *those without*. Here, a video sample along with its two descriptions is denoted as $(x, d_ 1, d_ 2)$. The outputs for the normal order input $(x, d_ 1, d_ 2)$ and the swapped order input $(x, d_ 2, d_ 1)$ are labeled $o_ {12}$ and $o_ {21}$, respectively. If a sample satisfies $o_ {12}=o_ {21}$, it is classified into the without order bias subset; otherwise, it is placed in the with order bias subset. We then report the recognition performance for these two subsets, with the results shown in Table 2. Experimental results reveal that the subset without order bias achieves higher recognition performance. In other words, **correctly predicted samples are less likely to exhibit order bias, whereas wrongly predicted samples show a higher likelihood of order bias.** This phenomenon occurs because when an MLLM fails to understand the emotions in certain challenging samples, it tends to guess emotion preferences randomly, leading to both lower recognition performance and a higher probability of order bias. **Therefore, we conclude that order bias is related to the MLLM’s emotional understanding capabilities and the difficulty of the samples.**
> >
> > #### *Table 2. Recognition performance comparison: subsets with vs. without order bias. This table takes Qwen2.5-Omni as the MLLM.*
> >
> > | Prompting Strategy | subset w/ order bias | subset w/o order bias |
> > |--------------------|----------------------|------------------------|
> > | S1                 | 0.46                 | 0.68                   |
> > | S2                 | 0.49                 | 0.70                   |
> > | S3                 | 0.33                 | 0.65                   |
> > | S4                 | 0.45                 | 0.70                   |
> >
> > *Q4: The paper designs four prompting strategies (S1-S4) with increasing reasoning chain complexity, but does not clarify why external LLMs improve performance in S3/S4. It only attributes it to "MLLMs may have limited language understanding," but does not verify whether the improvement comes from the external LLM’s stronger language reasoning ability or the multi-step reasoning’s error reduction.*
> >
> > A4: To understand why external LLMs improve performance, we visualize the internal outputs of S1$\sim$S4, as shown in Figures 21$\sim$23 (in the updated PDF). In Figure 21, we observe that S1/S2 struggle with instruction-following, particularly for complex prompts. Specifically, for the first step in S2, a short prompt asking the model to generate emotion descriptions yields results, even though the descriptions contain hallucinations. For the second step in S2, we use a relatively long and complex prompt, but the model fails to follow the instructions and instead continues focusing on description generation. These findings indicate that Video-ChatGPT has difficulty comprehending complex prompts. The root cause is that current MLLM typically begins with a pretrained LLM, followed by additional multimodal fine-tuning. While this enables the model to handle other modalities, such training may weaken the LLM's text understanding capabilities. In contrast, Figures 22 and 23 (in the updated PDF) show that by using an external LLM in S3/S4, we can generate outputs that align with our instructions. **Therefore, the performance improvement in S3/S4 is due to the external LLM’s stronger language understanding ability.**

---

### Official Review · Reviewer_ZiTv · 2025-11-06

**Soundness:** 3
**Presentation:** 3
**Contribution:** 3
**Rating:** 8
**Confidence:** 4

**Summary:**

This paper explores an interesting topic of human emotion preference, aiming to determine whether large language models can understand human preferences on different emotional representations. To this end, they construct EmoPrefer and introduce a new dataset (EmoPrefer-Data) and benchmark (EmoPrefer-Bench), supported by extensive experiments that reveal the upper-bound performance of current models. This work not only provides evaluation metrics for descriptive emotions but also advances future research toward developing more emotion-intelligent LLMs.

**Strengths:**

1.This paper proposes EmoPrefer, dedicated to human emotion preference.

2.This paper introduces a new dataset and a new benchmark, laying the foundations for this field.

3.This paper conducts extensive experiments, revealing the upper-bound performance of current models, and proposes techniques to enhance the performance on emotion preference prediction.

4.This paper has promising applications in descriptive emotion understanding and MLLM training.

**Weaknesses:**

1.	This paper aims to investigate whether MLLMs can replace humans in decoding emotion preferences. Besides experiments on EmoPrefer-Data, it would be beneficial to further discuss the relationship between MLLMs and humans in practical applications (mentioned in Figure 12).

2.	For EmoPrefer-Data, the authors primarily use samples with unanimous preference annotations. To better reveal human preferences, an analysis of which descriptions humans prefer most—considering factors such as emotion richness, description length, and modality coverage—would be necessary.

3.	The paper employs win/lose/tie counts and the Bradley-Terry algorithm to derive final rankings. To the best of our knowledge, alternative approaches exist—such as using binary 1/0 scores for win/lose without detailed counts. A discussion on why this ranking method was chosen in this paper should also be provided.

4.	Figure 10 shows that recognition performance is highly dependent on the choice of k. An explanation for this phenomenon would strengthen the analysis.

5.	Figure 7 indicates that the larger LLM (Qwen3-14B) underperforms compared to the smaller LLM (Qwen2.5-7B). This unexpected finding needs further explanation.

6.	The paper evaluates MLLMs in human emotion decoding under a zero-shot setup, with current models achieving around 70% recognition performance. While the focus of this paper is on evaluation, it would be valuable to discuss potential solutions for improving model performance in this task. Such discussion should be included in the future work.

**Questions:**

See wkes

---

> ### Author Response · Authors · 2025-11-21
> **Author Response (Part1)**
>
> Thanks for your support of EmoPrefer, including your recognition of our dataset, benchmark, and insights. We also appreciate your valuable suggestions.
>
> *Q1: This paper aims to investigate whether MLLMs can replace humans in decoding emotion preferences. Besides experiments on EmoPrefer-Data, it would be beneficial to further discuss the relationship between MLLMs and humans in practical applications (mentioned in Figure 12).*
>
> A1: Thanks for your comments. We analyze the correlation between MLLMs and humans using the examples mentioned in Figure 12. Specifically, we first recruit multiple human annotators to perform preference annotations. Then, we examine this correlation from two perspectives: *model-pair similarity* and *ranking similarity*.
>
> **(1) Model-pair Similarity.** Taking the comparison between $m_i$ and $m_j$ as an example. We denote the counts of wins, losses, and ties between $m_i$ and $m_j$ for human annotators as $\text{win}_ {ij}$, $\text{loss}_ {ij}$, and $\text{tie}_ {ij}$, and those for MLLMs as $\hat{\text{win}}_ {ij}$, $\hat{\text{loss}}_ {ij}$, and $\hat{\text{tie}}_ {ij}$. To evaluate the similarity between humans and MLLMs, we first convert these counts into three categories:
>
> \begin{equation}
>     {y}_ {ij} = \begin{cases}
>     \text{win},  & {\text{win}}_ {ij} > {\text{loss}}_ {ij}  \\\\
>     \text{lose}, & {\text{win}}_ {ij} < {\text{loss}}_ {ij}  \\\\
>     \text{tie},  & {\text{win}}_ {ij} = {\text{loss}}_ {ij}
>     \end{cases}
> \end{equation}
>
>
> \begin{equation}
>     \hat{y}_ {ij} = \begin{cases}
>     \text{win},  & \hat{\text{win}}_ {ij} > \hat{\text{loss}}_ {ij}  \\\\
>     \text{lose}, & \hat{\text{win}}_ {ij} < \hat{\text{loss}}_ {ij}  \\\\
>     \text{tie},  & \hat{\text{win}}_ {ij} = \hat{\text{loss}}_ {ij}
>     \end{cases}
> \end{equation}
>
> Figure 19 (in the updated PDF) presents the model-pair comparison results for humans and MLLMs. We then compute the similarity score between them:
>
> \begin{equation}
>     \text{Similarity} = \frac{\sum_ {i=1}^M\sum_ {j=i+1}^{M}\mathbb{I}[y_ {ij}=\hat{y}_ {ij}]}{\sum_ {i=1}^M\sum_ {j=i+1}^{M}1},
> \end{equation}
>
> where $\mathbb{I}(\cdot)$ is the indicator function. Experimental analysis reveals an 80.95\% agreement rate, indicating strong alignment between humans and MLLMs.
>
>
> **(2) Ranking Similarity.** Using the Bradley-Terry algorithm, we can derive model-level rankings from pairwise comparisons. In this section, we further evaluate the *ranking similarity* between MLLMs and humans. Specifically, we use Spearman’s rank correlation coefficient $\rho$ to quantify this similarity. Let $\text{Rank}_ A(\cdot)$ and $\text{Rank}_ B(\cdot)$ be two ranking functions. For each model $m_i$, its rank in the two rankings is denoted as $\text{Rank}_ A(m_ i)$ and $\text{Rank}_ B(m_ i)$. Spearman's $\rho$ is computed as:
>
> \begin{equation}
>     \rho=1-\frac{6\sum_{i=1}^M\left[\text{Rank}_A(m_i)-\text{Rank}_B(m_i)\right]^2}{M(M^2-1)}.
> \end{equation}
>
> This coefficient $\rho$ ranges from -1 to 1, with higher values indicating stronger positive correlation. Through experimental analysis, we observe that $\rho$ reaches 0.8571, a relatively high score, indicating good consistency between human and MLLM in decoding emotion preferences.
>
>
> *Q2: For EmoPrefer-Data, the authors primarily use samples with unanimous preference annotations. To better reveal human preferences, an analysis of which descriptions humans prefer most—considering factors such as emotion richness, description length, and modality coverage—would be necessary.*
>
> A2: Thanks for your suggestions. As shown in Tables 1 and 2, we examine the correlation between human preference with emotion richness and modality richness. These results reveal a general trend: *humans tend to prefer descriptions with more emotional words and a greater variety of modalities.* However, a significant proportion of exceptions exist. For instance, in some cases, humans favor descriptions with fewer emotional words and fewer modalities. **These findings suggest that human preference is also tied to the alignment of emotional descriptions with the video content, rather than simply prioritizing the richness of emotions or modalities.**
>
>
> #### *Table1. Correlation between human preference and emotion richness*
>
> | Human Preference         | Percentage (%) |
> |--------------------------|----------------|
> | More emotions            | 52.58          |
> | Fewer emotions           | 26.82          |
> | Same number of emotions  | 20.60          |
>
> #### *Table2. Correlation between human preference and modality richness*
>
> | Human Preference        | Percentage (%) |
> |-------------------------|----------------|
> | More modalities         | 23.80          |
> | Fewer modalities        | 11.37          |
> | Same number of modalities | 64.83          |

---

> ### Author Response · Authors · 2025-11-21
> **Author Response (Part2)**
>
> *Q3: The paper employs win/lose/tie counts and the Bradley-Terry algorithm to derive final rankings. To the best of our knowledge, alternative approaches exist—such as using binary 1/0 scores for win/lose without detailed counts. A discussion on why this ranking method was chosen in this paper should also be provided.*
>
> A3: Good question. Yes, there are many ways to derive rankings from pairwise comparison results.
>
> **(1) Our method.** Suppose there are $M$ models, each denoted as $m_i$. In this algorithm, each model $m_i$ is associated with a positive parameter $\theta_ i$, which quantifies its relative advantage. The probabilities that model $m_ i$ wins, loses, or ties against model $m_ j$ are defined as follows:
>
> \begin{equation}
>     P(m_ i>m_ j) = \frac{\theta_ i}{\theta_ i+\theta_ j+\beta},\\;P(m_ i<m_ j) = \frac{\theta_ j}{\theta_ i+\theta_ j+\beta},\\;P(m_ i \sim m_ j) = \frac{\beta}{\theta_ i+\theta_ j+\beta},
> \end{equation}
>
> where $\beta$ controls the likelihood of ties. The parameters $\theta$ can then be estimated by maximizing the following likelihood function. Models with higher $\theta$ values are ranked higher.
>
> \begin{equation}
>     \max \mathcal{L}(\theta) = \prod_ {i<j}\left(\frac{\theta_ i}{\theta_ i + \theta_ j+\beta}\right)^{\mathbf{\text{win}}_ {ij}}\left(\frac{\theta_ j}{\theta_ i + \theta_ j+\beta}\right)^{\text{lose}_ {ij}}\left(\frac{\beta}{\theta_ i + \theta_ j+\beta}\right)^{\text{tie}_ {ij}},
> \end{equation}
>
> where $\text{win}_ {ij}$, $\text{lose}_ {ij}$, and $\text{tie}_ {ij}$ denote the numbers of wins, losses, and ties between model $m_ i$ and model $m_ j$, respectively.
>
> **(2) Alternative approach.** In addition to our method, there are several alternative approaches. For example, the one you mentioned. Specifically, we can estimate the parameters $\theta$ using the following likelihood function:
>
> \begin{equation}
>     \max \mathcal{L}(\theta) = \prod_ {i<j}\left(\frac{\theta_ i}{\theta_ i + \theta_ j}\right)^{\mathbb{I}[\text{win}_ {ij}>\text{lose}_ {ij}]}\left(\frac{\theta_ j}{\theta_ i + \theta_ j}\right)^{\mathbb{I}[\text{win}_ {ij}<\text{lose}_ {ij}]},
> \end{equation}
>
> where we use binary 1/0 scores to represent wins and losses, without incorporating detailed numbers or considering tie cases.
>
> **(3) Discussion.** Compared to our method, this alternative approach is more susceptible to the influence of borderline cases. For instance, under this alternative method, the conclusions drawn from $\text{win}_ {ij}=49$ and $\text{lose}_ {ij}=51$ are entirely different from those drawn from $\text{win}_ {ij}=51$ and $\text{lose}_ {ij}=49$. In contrast, our method is more robust to such borderline scenarios. To support this claim, we conduct additional experiments on the practical application mentioned in Figure 12. We observe that the rankings generated by our method achieve a Spearman’s $\rho$ of 0.8571 when compared to human-based rankings, whereas the alternative method drops to 0.3214. These results demonstrate that our method is more consistent with human judgments.
>
>
> *Q4: Figure 10 shows that recognition performance is highly dependent on the choice of k. An explanation for this phenomenon would strengthen the analysis.*
>
> A4: Thanks for your comments. As shown in Figure 10, an inappropriate choice of $k$ can negatively affect the effectiveness of model-based crowdsourcing. Specifically, selecting a large $k$ may incorporate predictions from poor-performing models, thereby introducing additional noise. Conversely, choosing a small $k$ may result in the exclusion of useful information from well-performing models, thus limiting the overall effectiveness of model-based crowdsourcing. Consequently, the value of $k$ impacts the recognition performance.
>
>
> *Q5: Figure 7 indicates that the larger LLM (Qwen3-14B) underperforms compared to the smaller LLM (Qwen2.5-7B). This unexpected finding needs further explanation.*
>
> A5: This phenomenon indicates that larger models do not necessarily guarantee better performance in preference prediction. Our findings align with recent studies [1, 2], which demonstrate that some 72B models perform worse than well-designed 7B models in preference prediction. Beyond model size, performance depends on multiple factors, including training data, model architecture, and post-training strategies.
>
> [1] Zhang, Yi-Fan, Haihua Yang, Huanyu Zhang, Yang Shi, Zezhou Chen, Haochen Tian, Chaoyou Fu et al. "BaseReward: A Strong Baseline for Multimodal Reward Model." arXiv preprint arXiv:2509.16127 (2025).
>
> [2] Zhang, Yi-Fan, Xingyu Lu, Xiao Hu, Chaoyou Fu, Bin Wen, Tianke Zhang, Changyi Liu et al. "R1-reward: Training multimodal reward model through stable reinforcement learning." arXiv preprint arXiv:2505.02835 (2025).

---

> ### Author Response · Authors · 2025-11-21
> **Author Response (Part3)**
>
> *Q6: The paper evaluates MLLMs in human emotion decoding under a zero-shot setup, with current models achieving around 70\% recognition performance. While the focus of this paper is on evaluation, it would be valuable to discuss potential solutions for improving model performance in this task. Such a discussion should be included in future work.*
>
> A6: Thanks for your suggestions. To better address emotion preference decoding, we can make additional efforts in two directions in the future.
>
> **(1) Training-free methods.** This paper leverages model-based crowdsourcing to enhance recognition performance. Currently, we treat the prediction results of different models equally. In the future, we can introduce confidence scores during the fusion process. For example, we can assign more weight to high-confidence predictions and less weight to low-confidence predictions.
>
> **(2) Training-based methods.** We can train a dedicated model specifically for emotion preference decoding. Specifically, we can apply post-training techniques, such as supervised fine-tuning or reinforcement learning, to improve the model’s performance on this task.

---

### Author Response · Authors · 2025-12-01
**Global Response**

Dear AC, SAC, and PC,

Thank you all for taking the valuable time to review our paper. Below, we summarize the discussions during the review period and provide our responses to the reviewers’ comments.

*Reviewers ZiTv, y5v9, and evbf* gave positive feedback on our work (Rating: 8 or 6). Through additional experiments and further clarifications, we have thoroughly addressed all of their questions and concerns.

*Reviewer yfmT* gave a negative rating (Rating: 2), but with very low confidence (Confidence: 2), indicating limited familiarity with this field. The reviewer initially made several factual errors regarding the compared description pairs. We addressed these concerns from multiple perspectives, including the benefits of human–LLM collaboration in emotion description generation, the developmental history of descriptive emotions, and mainstream preference datasets in related domains. Following our response, the reviewer revised their rating upward, suggesting that we have adequately addressed their concerns.

*Reviewer uypZ* acknowledged the soundness, presentation, and contributions of our work and expressed interest in our work. However, the reviewer misunderstood the diversity of our dataset, which led to an initial rating of 4. In our rebuttal, we have demonstrated the diversity of the data sources used in the current dataset. Furthermore, we expanded the dataset and constructed a new one, which effectively addresses the reviewer’s concerns. We trust that, assuming no technical issues with OpenReview, reviewer uypZ will increase his rating.

We hope this summary helps facilitate your decision-making process. Thank you once again for your time and consideration.

Sincerely,

The Authors of Manuscript 9269

---

### Meta-Review · Area_Chair_zbCe · 2026-01-11

**Summary:**

Reviewers appreciated the novelty of the task and the new dataset. A number of the reviewers raised concerns mostly about some evaluation choices ανδ have pointed out that the dataset is not as diverse as it seems. Authors have responded satisfactorily to the concerns of the negative reviewers.

**Reviewer Concerns:**

The authors have addressed concerns about additional evaluation and have provided some additional insights based on the reviewers criticisms. The concerns of the relatively limited dataset in terms of diversity are only partially addressed. Indeed the scenarios are not as broad as on would expect and the front facing characters.

The points raised by the negative reviewers were not as strong as the positive comments of the psotove reviewers and it appears that the most negative reviewer had low confidence and misunderstood the paper originally as seen from their rebuttal response.

The criticism about the lack of diversity in terms of limited scenarios and mostly front-facing speakers is valid, but the authors partially address it making a point that the dataset is still useful and that these limitations make sense for this initial approach.

One limitation that was only tangentially raised by Rev ZiTv  concerns the limited discussion on practical applications. It is not trivial to me why evaluating whether MLLMs agree with humans on which MER description is better, might be useful for a real-world task. There might be applicable real-world tasks out there, but it's not obvious to me --> more discussion is needed. Understanding human emotion, and thus agreeing with humans in emotion annotation, is important but I'm not sure whether this binary choice between free-form LLM-generated text is a meaningful experimental scenario.

**Reviewer Scores:**

I believe all reviewers would have given a positive score as the authors have responded satisfactorily to the concerns of the negative reviewers.

---

### Decision · Program_Chairs · 2026-01-26

Accept (Poster)